# Tuning the Implicit Regularizer of Masked Diffusion Language Models: Enhancing Generalization via Insights from $k$-Parity

**Jianhao Huang**[1]   **Baharan Mirzasoleiman**[1]

## Abstract

Masked Diffusion Language Models have recently emerged as a powerful generative paradigm, yet their generalization properties remain understudied compared to their autoregressive counterparts. In this work, we investigate these properties within the setting of the $k$-parity problem (computing the XOR sum of $k$ relevant bits), where neural networks typically exhibit grokking—a prolonged plateau of chance-level performance followed by sudden generalization. We theoretically decompose the Masked Diffusion (MD) objective into a Signal regime which drives feature learning, and a Noise regime which serves as an implicit regularizer. By training nanoGPT using MD objective on the $k$-parity problem, we demonstrate that MD objective fundamentally alters the learning landscape, enabling rapid and simultaneous generalization without experiencing grokking. Furthermore, we leverage our theoretical insights to optimize the distribution of the mask probability in the MD objective. Our method significantly improves perplexity for 50M-parameter models and achieves superior results across both pre-training from scratch and supervised fine-tuning. Specifically, we observe performance gains peaking at $8.8\%$ and $5.8\%$, respectively, on 8B-parameter models, confirming the scalability and effectiveness of our framework in large-scale masked diffusion language model regimes.

## 1. Introduction

Masked Diffusion Language Models (MDLMs) (Lou et al., 2024; Gong et al., 2025a; Nie et al., 2025a;b; Ye et al., 2025) have recently emerged as a powerful alternative to the dominant autoregressive paradigm, achieving performance levels that rival standard Autoregressive Models (ARMs) (OpenAI, 2023; Liu et al., 2024; Comanici et al., 2025; Yang et al., 2025). Unlike ARMs, which are guided by next-token prediction, MDLMs utilize a training framework where tokens are independently masked at a ratio $t \sim \mathcal{U}[0, 1]$, requiring the model to reconstruct the original data from its corrupted version. Interestingly, recent work demonstrates that MDLMs can outperform ARMs in low-data regimes (Ni et al., 2025a) and in the absence of weight decay (Ni et al., 2025b), suggesting a natural superiority in generalization (see Section 2). However, the role of the MDLM objective in steering optimization toward generalizable solutions has yet to be rigorously characterized.

To answer this quesiton, we study a simplified setting—a Transformer trained with a masked diffusion loss on the $k$-parity problem—to uncover the fundamental mechanisms that drive MDLM learning. The $k$-parity problem typically presents a formidable challenge for standard training paradigms, which often fall into a "grokking" trap (Power et al., 2022): a prolonged plateau of chance-level performance followed by sudden, late-stage generalization.

We demonstrate that the masked diffusion (MD) objective inherently circumvents this challenge. By analytically decomposing the objective, we reveal that the loss naturally partitions into a **Signal Regime** and a **Noise Regime**:

$$\mathcal{L}_{\text{eff}}(\theta) \approx P_S \underbrace{\mathbb{E}_S[\|f_\theta(\tilde{\mathbf{z}}) - f^*\|^2]}_{\text{Task Signal}} + P_N \underbrace{\mathbb{E}_N[\|f_\theta(\tilde{\mathbf{z}})\|^2]}_{\text{Implicit Regularization}} .$$

Intuitively, the Noise Regime arises when the masking process is either too aggressive, obscuring the bits necessary to compute the parity, or when it only masks irrelevant positions, leaving the target bit untouched. In either case, the resulting sample becomes information-theoretically unidentifiable. Because these unidentifiable examples do not provide sufficient information to reconstruct the target, the objective forces the model to minimize its output norm on these inputs. This mechanism serves as an implicit regularizer that prevents the model from falling into pure memorization,

[1]Department of Computer Science, University of California, Los Angeles. Correspondence to: Jianhao Huang <jh-huang2025@cs.ucla.edu>.

*Proceedings of the 43$^{rd}$ International Conference on Machine Learning*, Seoul, South Korea. PMLR 306, 2026. Copyright 2026 by the author(s).

thereby promoting robust algorithmic generalization.

Building on this structural insight, we hypothesize that the standard practice of uniform mask sampling ($t \in [0,1]$) is suboptimal for natural language since the extreme ends of the process ($t \approx 0$ and $t \approx 1$) may provide vanishingly small gradient signals or redundant regularization. We propose a **Signal-Rich Mask Sampling** strategy that concentrates training on the most informative masking intervals. We validate our framework across multiple scales from toy level to pretraining and fine-tuning 8B-parameter models.

We thus **summarize our primary contributions as follows:**

- **Theoretical Discovery of Implicit Regularization.** Using the $k$-parity problem as an analytical framework, we provide a formal decomposition of the MD objective into a *Signal Regime* and a *Noise Regime*. We prove that the latter functions as a built-in implicit regularizer by penalizing model predictions on information-theoretically obscured inputs, a mechanism that eliminates the generalization delay typical of standard supervised objectives.

- **Empirical Verification on the Parity Case.** We demonstrate that the MD objective fundamentally alters the learning dynamics of discrete algorithmic tasks, enabling near-instant generalization on $k$-parity benchmarks. Notably, MD escapes the "grokking" phenomenon entirely. To the best of our knowledge, we are the **first** to identify that masked diffusion can achieve such rapid generalization on parity.

- **Generalization of Theoretical Insights to Large-Scale Models.** We demonstrate that the principles derived from the $k$-parity case translate effectively to real-world language modeling and reasoning. We validate this transfer across two distinct scales. (i) At 50M scale, we show that tuning the masking range to concentrate on high-signal regimes significantly improves perplexity on the `WikiText` dataset. (ii) We demonstrate that our signal-rich distribution remains robust at the 8B-parameter scale. Our approach achieves up to $8.8\%$ improvement in pre-training. In supervised fine-tuning (SFT), optimizing noise intervals yields gains of up to $5.8\%$ on discriminative tasks and $3.4\%$ on complex generative reasoning.

## 2. Related Work

**Masked Diffusion Language Models.** Masked Diffusion Language Models (MDLM) (Lou et al., 2024; Gong et al., 2025a; Nie et al., 2025a;b; Ye et al., 2025) have recently emerged as a powerful alternative to the dominant autoregressive paradigm, achieving performance levels that rival standard Autoregressive Models (ARMs). Re-

cent scaling-law analysis (Ni et al., 2025a;b) demonstrate a unique compute-data tradeoff: while MDLMs are more data-intensive under fixed compute budgets, they outperform ARMs when training longer on small data.

A particularly striking property of MDLM is its intrinsic robustness to overfitting, which manifests across two distinct settings. First, in low data-budget settings, MDLMs remain largely unaffected by data repetition, whereas ARMs exhibit severe degradation in validation loss and benchmark performance (Ni et al., 2025a). Second, MDLMs maintain stable performance even without weight decay (Ni et al., 2025b), while removing weight decay typically compromises the generalization of ARMs.

Theoretically, Shi et al. (2024) simplified the MDLM objective into a weighted integral of cross-entropy losses. Sahoo et al. (2024) introduced a simplified, Rao-Blackwellized objective that takes the form of a weighted average of masked language modeling losses (Devlin et al., 2019). Ou et al. (2025) established its equivalence to the expected negative log-likelihood of any-order autoregressive models (AO-ARMs). While these studies clarify the structural form of the objective, they do not explain the superior generalization capabilities of the MDLM objective. We bridge this gap by analytically decomposing the MD objective, demonstrating that it contains an inherent implicit regularizer that penalizes the model's output on unidentifiable inputs, thereby preventing memorization and facilitating rapid generalization.

We provide a detailed discussion for more related works of MDLM in Section A.

**Grokking** is first discovered by Power et al. (2022) on a set of small algorithmic reasoning tasks. The intriguing phenomenon inspired follow-up works proposing different explanations. Merrill et al. (2023) describe it as a competition between generalization and memorization circuits, while Kumar et al. (2023) frame it as a transition from "lazy" to "rich" learning regimes. Additionally, recent research has analyzed grokking dynamics within specific contexts, such as clustering tasks (Xu et al., 2024), linear networks (Dominé et al., 2025). Within the context of group arithmetic, Tian (2025) highlights weight decay as the critical mechanism for triggering the transition from memorization to algorithmic solutions.

## 3. Preliminaries

In this section, we establish the formal framework required to analyze the learning dynamics of masked diffusion. We first introduce the $k$-parity problem, which serves as a theoretical testbed for the following analysis. We then define the architecture of a one-layer Transformer and formalize the stochastic training objective.

**Notation** We use $[T]$ to denote the set $\{1, 2, ..., T\}$. Scalars are in lower-case unbolded letters ($y, \alpha$, etc.). Matrices and vectors are denoted in upper-case bold letters ($\boldsymbol{W}, \boldsymbol{V}$, etc.) and lower-case bold letters ($\boldsymbol{x}, \boldsymbol{w}$, etc.), respectively. $\boldsymbol{W}_{[i,j]}, \boldsymbol{W}_{[i,:]}, \boldsymbol{W}_{[:,j]}$ respectively denotes the $(i, j)$-th entry, $i$-th row, and $j$-th column of the matrix $\boldsymbol{W}$. $\boldsymbol{W}_{[:,-1]}$ means the last column of the matrix $\boldsymbol{W}$. The notation $\boldsymbol{W}_{ij}$ denotes block matrices/vectors on the $i$-th row and $j$-th column according to context. We use $\mathbb{1}\{\cdot\}$ to denote the indicator function, use $\odot$ to denote the Hadamard product and use $\dagger$ to denote the Moore-Penrose pseudoinverse.

## 3.1. Parity

To move beyond empirical observations and isolate the mechanistic cause of MDLM's robustness, we adopt the parity task as our theoretical playground. Parity is a well-studied theoretical benchmark in learning theory (Daniely & Malach, 2020; Abbe et al., 2022; 2023; Barak et al., 2022; Edelman et al., 2023; Kou et al., 2024; Kim & Suzuki, 2025; Wen et al., 2025), frequently utilized to analyze the ability of neural architectures to extract low-rank signal from high-dimensional noise. Furthermore, the task constitutes a canonical case of grokking (Barak et al., 2022; Merrill et al., 2023; Bautiste et al., 2024; Prieto et al., 2025; Pasand & Dohmatob, 2025). By stripping away the linguistic complexities of natural language, parity provides a controlled environment to determine whether the MD objective inherently suppresses suboptimal memorization trajectories in favor of structural learning.

**Definition 3.1** ($(n, k)$ Parity Task). Let $\mathcal{S} \subset [n]$ be a secret set of indices with $|\mathcal{S}| = k$. For an input $\boldsymbol{x}$ drawn uniformly from $\{-1, 1\}^n$, the label $y_{\boldsymbol{x}}$ is defined as the product of the bits in $\mathcal{S}$: $y_{\boldsymbol{x}} = \prod_{j \in \mathcal{S}} x_j$.

**Definition 3.2** (Full Sequence $\boldsymbol{x}'$). For training, the $n$ input bits and the parity label are concatenated into a single sequence $\boldsymbol{x}' \in \{-1, 1\}^{n'}$ of length $n' = n + 1$

$$\boldsymbol{x}' = (x'_1, x'_2, \ldots, x'_n, x'_{n+1})$$

where $x'_j = x_j$ for $j \in [n]$, and $x'_{n+1} = y_{\boldsymbol{x}}$. Define $\mathcal{S}' = \mathcal{S} \cup \{n'\}$ to be the secret set of the sequence $\boldsymbol{x}'$.

We consider a finite training dataset $\mathcal{D}' = \{\boldsymbol{x}'_i\}_{i=1}^N$ of size $N$, where each example $\boldsymbol{x}'_i$ is independently generated as described above.

## 3.2. One-Layer Transformer

We consider a one-layer transformer equipped with single-head self-attention (SSA) and a feedforward network (FFN). Following Zhang et al. (2024); Nichani et al. (2024); Huang et al. (2025), we consolidate the projection matrix in FFN and the value matrix in SSA into a single matrix $\boldsymbol{W}$, and merge the key and query matrices into $\boldsymbol{A}$.

**Definition 3.3** (One-Layer Transformer). Let $\boldsymbol{v} \in \mathbb{R}^D$ and $\boldsymbol{W} \in \mathbb{R}^{D \times d}$ be the FFN matrices and $\boldsymbol{A} \in \mathbb{R}^{d \times d}$ be the attention matrix. Let $\sigma$ be the element-wise activation function such as ReLU. Define the parameter $\theta := \{\boldsymbol{v}, \boldsymbol{W}, \boldsymbol{A}\}$. The one-layer transformer $\mathrm{TF}_\theta$ operates on $\boldsymbol{X} \in \mathbb{R}^{d \times n}$ by

$$\mathrm{TF}_\theta(\boldsymbol{X}) = \boldsymbol{v}^\top \sigma\big(\boldsymbol{W}\boldsymbol{X}\mathrm{softmax}\big(\boldsymbol{X}^\top \boldsymbol{A}\boldsymbol{X}\big)\big) \in \mathbb{R}^{1 \times n}.$$

We remark that the softmax function is applied colume-wise.

## 3.3. Masked Diffusion Loss

**Definition 3.4** (Stochastic Masking). For a training sample $x'$, a mask probability $t$ is sampled from a uniform distribution, $t \sim U[t_0, t_1]$. A mask vector $\boldsymbol{m} \in \{0, 1\}^{n'}$ is then sampled, where each component $m_j$ is an independent Bernoulli trial $m_j \sim \mathrm{Bernoulli}(t)$.

We define the set of *masked* indices as $M_{\boldsymbol{m}} = \{j \mid m_j = 1\}$. Given $\boldsymbol{x}'$ and $\boldsymbol{m}$, define the corrupted sample

$$\tilde{\boldsymbol{x}}(\boldsymbol{x}', \boldsymbol{m}) = \boldsymbol{x}' \odot \sim \boldsymbol{m}.$$

**Definition 3.5** (Token Embeddings). The model operates in an embedding space $\mathbb{R}^d$ where $d = 3n' = 3(n + 1)$. This space provides a unique, orthogonal one-hot embedding for each token value $b \in \{-1, 0, 1\}$ (where '0' represents a mask token) at each position $j \in [n']$.

The embedding for value $b$ at position $j$ is the one-hot vector $\boldsymbol{e}_{n'b+j} \in \mathbb{R}^d$. Given $\boldsymbol{x}'$ and mask $\boldsymbol{m}$, We denote the input as

$$\widetilde{\boldsymbol{X}}(\tilde{\boldsymbol{x}}) = \big(\cdots \quad \boldsymbol{e}_{n'\tilde{x}_j+j} \quad \cdots\big) \in \mathbb{R}^{d \times n'}.$$

We denote $\widetilde{\boldsymbol{X}}(\tilde{\boldsymbol{x}})$ simply as $\widetilde{\boldsymbol{X}}$ when the context is clear.

Following recent empirical works (Nie et al., 2025b; Ye et al., 2025; Ni et al., 2025b), we consider full batch gradient flow dynamics over the mean squared loss computed only on the masked tokens.

**Definition 3.6** (Training Loss Function). The model is trained to predict the original values of the masked tokens. Given $t_i$ and $\boldsymbol{m}_i$ sampled, the loss for a single sample $\boldsymbol{x}'_i$ is defined as:

$$\ell(\theta|\boldsymbol{x}'_i, t_i, \boldsymbol{m}_i) = \frac{1}{2t_i} \sum_{j \in M_{t_i}} \Big(f(\widetilde{\boldsymbol{X}}_i)_{[:,j]} - x'_j\Big)^2.$$

The overall training objective is the expected loss over the data, mask probability, and mask distributions:

$$\mathcal{L}(\theta) = \mathbb{E}_{\boldsymbol{x}' \in \mathcal{D}', t \sim U[t_0, t_1], \boldsymbol{m}} \left[\ell(\theta|\boldsymbol{x}', t, \boldsymbol{m})\right]. \quad (1)$$

## 3.4. Evaluation

To evaluate the model's ability to recover the underlying logic of the parity task, we test its performance on a specific,

non-stochastic evaluation mask $\mathbf{m}_{\text{eval}}$. In this setting, the model is provided with all input bits $x_1, \ldots, x_n$, but the final parity bit $x'_{n'} = y_{\boldsymbol{x}}$ is masked, such that $M_{\text{eval}} = \{n'\}$. The evaluation input is constructed as

$$\widetilde{\boldsymbol{x}}_{\text{eval}} = \widetilde{\boldsymbol{x}}(\boldsymbol{x}' \odot \sim \boldsymbol{m}_{\text{eval}}).$$

The model's prediction is considered correct if the sign of the output at masked position matches the true parity bit $y_{\boldsymbol{x}}$.

# 4. Understanding Masked Diffusion through Parity

In this section, we investigate the learning process of masked diffusion using the parity task. We first demonstrate that the Transformer's attention mechanism is not necessary for this task, allowing us to reduce the architecture to a two-layer fully connected neural network. We then provide a theoretical decomposition of the training objective into Signal and Noise regimes. Finally, we provide empirical evidence using a nanoGPT implementation.

## 4.1. Structure Reduction

While previous theoretical investigations of the parity task have primarily utilized 2-layer multi-layer perceptron (MLP) (Daniely & Malach, 2020; Abbe et al., 2022; 2023; Barak et al., 2022; Edelman et al., 2023; Kou et al., 2024), it is not immediately clear whether such simplifications hold for the MD objective. Unlike standard supervised learning, the objective involves a stochastic masking process that may rely on the Transformer's attention mechanism to aggregate information across positions.

To resolve this, we conduct a preliminary ablation by reducing the attention mechanism to **uniform attention**. We empirically find that the model still remains trainable and shows no sign of grokking (refer to Section F.1). This discovery confirms that the core generalization dynamics of masked discrete diffusion, in this context, are independent of the attention mechanism.

Consequently, we reduce the transformer to a 2-layer MLP for our theoretical analysis. In the uniform attention case, the softmax operation simplifies to a constant uniform distribution $1/n'$ across all positions. We absorb the constant into parameters. Then, the model output $\text{TF}_\theta(\widetilde{\boldsymbol{x}})$ at the each position is the same and becomes a function of the global sum of input embeddings. We define the aggregated input vector $\tilde{\boldsymbol{z}}$ as

$$\tilde{\boldsymbol{z}} = \sum_{j=1}^{n'} \mathbf{e}_{n'\tilde{x}_j + j} \in \mathbb{R}^d.$$

The reduced model with parameters $\theta = \{\boldsymbol{v}, \boldsymbol{W}\}$ then oper-

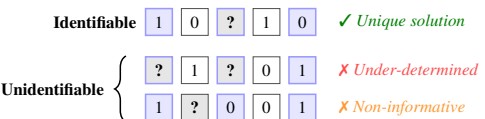

*Figure 1.* Identifiability of masking patterns in the $(n, k) = (4, 2)$ parity task. Blue cells indicate the secret set $\mathcal{S}$ and grey cells indicate the masked positions.

ates on this aggregate vector as

$$f_\theta(\tilde{\boldsymbol{z}}) = \boldsymbol{v}^\top \sigma(\boldsymbol{W}\tilde{\boldsymbol{z}}) \in \mathbb{R}.$$

## 4.2. Signal and Noise Regimes

To characterize the learning dynamics, we must distinguish between input configurations where the masked tokens are information-theoretically identifiable (Signal) and those where they are statistically independent of the visible tokens (Noise) (see Figure 1).

The stochastic masking process induces a distribution over the aggregated input vectors $\tilde{\boldsymbol{z}}$. We partition the space of inputs into two disjoint regimes based on the mask $\boldsymbol{m}$.

**Definition 4.1** (Signal and Noise Regimes). We classify mask configurations $\boldsymbol{m}$ into two regimes based on the extended secret set $\mathcal{S}'$:

(i) **Signal Regime** $\mathcal{R}_S = \{\boldsymbol{m} \mid |M_{\boldsymbol{m}} \cap \mathcal{S}'| = 1\}$, where the masked token $x'_j$ is determined by unmasked tokens in $\mathcal{S}'$.

(ii) **Noise Regime** $\mathcal{R}_N = \{\boldsymbol{m} \mid |M_{\boldsymbol{m}} \cap \mathcal{S}'| \neq 1\}$, where all masked token is unidentifable.

Before analyzing how a model learns, we must ensure the parity function is uniquely determined by the available data. Unlike standard parity, the model only observes corrupted samples. We provide the following information-theoretic bound:

**Theorem 4.2** (Information-Theoretic Identifiability). *Let $\mathcal{D}'$ be a dataset of $N$ samples subjected to the stochastic masking process in Definition 3.4. To ensure that the secret set $\mathcal{S}$ is uniquely identifiable from the corrupted dataset with probability at least $1 - \delta$, the number of samples $N$ must satisfy:*

$$N \geq \frac{4 \log(4n/\delta)}{\mathbb{E}[t] \cdot (\mathbb{E}[(1-t)^k])^2}. \tag{2}$$

*where $k = |\mathcal{S}|$ is the size of the secret set (excluding the parity bit itself).*

Please refer to Section B.1 for the whole proof. This result establishes the data floor required for the algorithm to succeed. With this identifiability guaranteed, We now analyze the landscape of the MD objective to explain how it facilitates rapid generalization on the parity task.

## 4.3. Effective Loss and Energy Landscape

We now analyze the gradient flow dynamics by projecting the finite sum loss onto the distribution of aggregated embeddings $\tilde{z}$.

**Theorem 4.3** (Effective Loss Decomposition). *Consider the training objective Equation* (1) *over a finite dataset $\mathcal{D}'$. For $m \in \mathcal{R}_S$, we define $f^*(\tilde{z}) = \frac{1}{|M_m|} x'_{M_m \cap \mathcal{S}'}$ to be the optimal target function. The loss is equivalent to:*

$$\mathcal{L}_{\text{eff}}(\theta) = P_S \mathbb{E}_{\tilde{z}|\mathcal{R}_S}\left[\frac{|M_m|}{2t}(f_\theta(\tilde{z}) - f^*(\tilde{z}))^2\right]$$
$$+ P_N \mathbb{E}_{\tilde{z}|\mathcal{R}_N}\left[\frac{|M_m|}{2t}f_\theta(\tilde{z})^2\right], \quad (3)$$

*where $P_S = (k+1)\mathbb{E}_{t \sim U[t_0, t_1]}\left[t(1-t)^k\right]$ and $P_N = 1 - P_S$. The expectations are taken over the empirical distribution of $\tilde{z}$ conditioned on the respective regimes.*

Please refer to Section C.1 for the whole proof. The decomposition in Equation (3) reveals that the MD objective functions as a composite loss with a built-in regularizer. While the first term (Signal Regime) drives the model to map identifiable masked patterns to their latent labels, the second term (Noise Regime) acts as an **implicit regularizer** on the model's output.

*Remark* 4.4 (Extension to the Cross-Entropy Loss). The same signal–noise structure carries over to the binary cross-entropy (CE) objective, confirming that the mechanism is not an artifact of the squared loss but is intrinsic to the masking process itself. We refer the reader to Section C.2 for the full derivation.

To characterize the mechanism of feature learning, we adopt the *lazy readout* assumption following Tian et al. (2023); Marion & Berthier (2023); Berthier et al. (2025); Tian (2025), assuming the linear readout $v$ converges to its optimal value $v^*(W)$ much faster than hidden weights $W$. Under this assumption, the optimization of the effective loss reduces to maximizing an energy function $E(W)$, which characterizes the landscape governing feature learning:

**Theorem 4.5** (Energy Landscape). *Consider the effective loss $\mathcal{L}_{\text{eff}}(v, W)$ defined in Equation* (3). *Let $h(\tilde{z}) = \sigma(W\tilde{z})$. Let the correlation vector $c(W)$ and the covariance matrix $\Sigma(W)$ be defined as*

$$c(W) = P_S \mathbb{E}_S\left[\frac{|M_m|}{2t}f^* h\right], \quad \Sigma(W) = \mathbb{E}\left[\frac{|M_m|}{2t}hh^\top\right].$$

*We assume the readout $v$ is at optimality for a fixed $W$. Then minimizing the effective loss is equivalent to maximizing the energy $E(W)$*

$$E(W) = c(W)^\top \Sigma(W)^\dagger c(W).$$

Please refer to Section D.1 for the whole proof. Since $E(W) \propto P_S^2$, the signal ratio $P_S$ acts as a dynamic gain factor. It governs the ascent velocity on the energy landscape, effectively scaling the learning rate of $W$ specifically in the direction of the target $f^*$.

While $P_N \to 0$ is practically impossible in masked diffusion, considering this limit reveals why the noise component is essential to maintain optimization.

**Corollary 4.6** (Feature Learning Collapse in Pure Signal Regime). *Consider the limit where training falls entirely into the Signal Regime ($P_N \to 0$). If the network is sufficiently expressive, the energy function saturates at a constant:*

$$E(W) \xrightarrow{P_N \to 0} \text{Const}.$$

Please refer to Section D.2 for the whole proof. Corollary 4.6 formalizes a critical failure mode where the gradients for the internal parameters $W$ vanish when the training falls entirely into the Pure Signal Regime ($P_N \to 0$). In this regime, a sufficiently expressive network allows the feature covariance $\Sigma(W)$ to perfectly adapt and neutralize the directional information contained in $c(W)$. Because the energy function $E(W)$ saturates at a constant value, the resulting gradient $\nabla_W E(W)$ becomes zero. This effectively halts the evolution of the feature extractor and prevents the model from learning more sophisticated representations.

The result presented here is consistent with the findings in Tian (2025), where it was shown that feature learning can collapse without proper regularization. In those cases, the model often settles into a lazy state where the internal weights stop receiving informative updates once the immediate training objective is met through simple correlation.

## 4.4. Optimal Mask Sampling

In this section, we derive the optimal sampling distributions for $t$ under two different optimization criteria: **Sample Complexity-Optimality**, which minimizes the data required for structure discovery, and **Signal-Optimality**, which maximizes the clarity of the learning gradient.

We first formalize the strategy for minimizing data requirements by leveraging the information-theoretic bounds established in our earlier analysis. By Theorem 4.2, we directly obtain the optimal strategy in terms of the best sample complexity lower bound.

**Corollary 4.7** (Sample Complexity-Optimal Masking Rate). *Assume the masking rate follows a uniform distribution $t \sim U[t_0, t_1]$. The sample complexity objective dictates the optimal strategy by minimizing the lowerbound in Equation* (2) *as*

*(i) Simple Dependencies ($k = 1$): Any uniform distribution is optimal provided the mean is $\mathbb{E}[t] = \frac{1}{3}$.*

*(ii) Complex Dependencies ($k > 1$): The optimal distribution is anchored at the lower boundary $t_0 = 0$. The upper bound $t_1 \in (0,1)$ is the root of:*

$$(2k+1)(1-t_1)^{k+1} - (2k+2)(1-t_1)^k + 1 = 0.$$

While Corollary 4.7 provides a bound on the data volume required for success, it does not account for the clarity of the gradient during the learning process. To address this, we consider a second criterion based on Theorem 4.3 and Theorem 4.5. This Signal-Optimal strategy aims to maximize the identifiable signal component $P_S$, effectively centering the training process on the masking rates that most clearly expose the underlying structure.

**Corollary 4.8** (Signal-Optimal Masking Rate). *Assume the masking rate follows a uniform distribution $t \sim U[t_0, t_1]$. The signal-noise ratio dictates the optimal strategy by maximizing $P_S$ as*

$$t_0 = t_1 = \frac{1}{k+1}.$$

Please see Section B.2 and Section E respectively for the whole proof. Notably, both objectives follow the functional form of $t(1-t)^C$. Consequently, the masking rate $t$ cannot be too small or too large, as the expression vanishes toward zero in both limits.

While both criteria provide theoretical instructions, we prioritize the signal-optimal strategy for our subsequent analysis. This choice reflects the fact that natural language is characterized by significant redundancy rather than a singular objective mapping, making signal maximization a more robust driver of learning efficiency and performance.

### 4.5. Experiments on $k$-parity

We conduct experiments on the $(n, k) = (20, 6)$ parity task using a standard Transformer architecture and cross-entropy loss to validate our theoretical framework. The model follows the nanoGPT implementation, incorporating self-attention, layer normalization, residual connections, and an MLP block. To isolate the effects of the diffusion objective, we set the weight decay $\eta = 0.1$ for all runs. The empirical results illustrated in Figure 2 yield several key observations that confirm the dynamics of the energy landscape.

**Elimination of Grokking and Simultaneous Generalization.** The blue curve, representing standard training, exhibits classic grokking behavior. While the model achieves $100\%$ training accuracy almost immediately, the validation accuracy remains at a baseline of $50\%$, indicating a failure to learn generalizable features in the early stage. In contrast, masked diffusion (purple and orange) facilitates near-simultaneous convergence of training and validation accuracy. This confirms that the MD objective provides

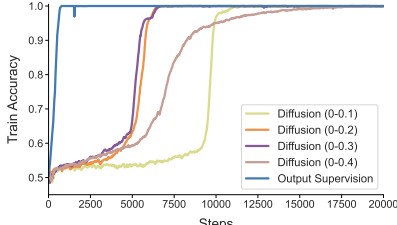

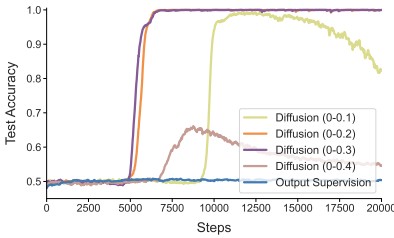

*Figure 2.* **Training and validation accuracy on $(n, k) = (20, 6)$ parity.** Standard training (blue, directly supervising $y_x$) exhibits classic grokking behavior; while training accuracy reaches $100\%$ rapidly, validation accuracy remains at chance level ($50\%$) for a prolonged period before eventually converging to 1 (see full trajectory in Section F.2). In contrast, Masked Diffusion (purple, orange) enables near-simultaneous generalization. The fastest convergence is achieved near the theoretically predicted optimal range ($\mathcal{U}[0, 0.246]$). Boundary cases ($\mathcal{U}[0, 0.1]$ and $\mathcal{U}[0, 0.4]$) exhibit instability and slower convergence due to excessive regularization.

an inherent regularization that boosts the feature learning process.

**Verification of the Signal-Optimal Range.** The orange ($\mathcal{U}[0, 0.2]$) and purple ($\mathcal{U}[0, 0.3]$) curves exhibit the fastest convergence speeds, significantly outperforming other configurations. This empirical observation aligns closely with our theoretical derivation, which suggests that for a fixed $t_0 = 0$, the optimal range resides near the $\mathcal{U}[0, 0.246]$ range where $P_S = 7\mathbb{E}_{t \sim U[0, t_1]}\big[t(1-t)^6\big]$ is maximized.

**Boundary Instability.** The erratic performance of the yellow ($\mathcal{U}[0, 0.1]$) and brown ($\mathcal{U}[0, 0.4]$) curves reflects the optimization instability caused by insufficient signal. In both cases, $P_S$ is lower than in the orange and purple configurations. This lack of signal diminishes the update magnitude for $W$, making the learning process more susceptible to sampling noise and slowing down overall convergence. Furthermore, excessive regularization drastically alters the loss landscape which obscures the true global minima and traps the model in suboptimal solutions.

## 5. Signal-Rich Mask Sampling in Language Modeling

We now translate these theoretical insights into practical improvements for generative language modeling. Our anal-

ysis of the $k$-parity problem reveals that regularization is a double-edged sword: while essential for generalization, excessive regularization distorts the optimization landscape, which not only impedes learning efficiency but also leads to suboptimal convergence. We argue that the $t \sim U[0,1]$ distribution in natural language modeling may impose an over-regularizing effect that similarly degrades both training speed and model performance. Consequently, we hypothesize that the optimal training regime shifts significantly toward maximizing signal intensity.

This shift suggests that the extrema of the standard uniform sampling range, $t \sim \mathcal{U}[0,1]$, contribute little to the underlying learning dynamics. At the lower bound ($t \to 0$), the task collapses into trivial reconstruction, yielding vanishing gradients that stall feature extraction. Conversely, at the upper bound ($t \to 1$), the input becomes information-theoretically void, forcing the model to predict marginal token distributions without the context necessary to learn inter-token dependencies.

To concentrate the model's capacity on the most informative samples, we propose restricting the training to a Signal-Rich window, $t \in [t_{\min}, t_{\max}]$. By bypassing the computationally inefficient boundaries, we ensure that gradient updates are consistently driven by meaningful context. Under this framework, the training objective becomes the cross-entropy loss restricted to the signal-rich interval. Specifically, given a sequence $\boldsymbol{x}_0$, we sample a time step $t \sim \mathcal{U}[t_{\min}, t_{\max}]$ to construct the masked sequence $\boldsymbol{x}_t$, training the model $p_\theta$ to minimize

$$\mathcal{L}(\theta) = -\mathbb{E}_{t,\boldsymbol{x}_0,\boldsymbol{x}_t}\left[\frac{1}{t}\sum_{i=1}^{L}\mathbb{1}\left[x_t^i = M\right]\log p_\theta\left(x_0^i \mid \boldsymbol{x}_t\right)\right].$$

Crucially, while our training objective is localized to the signal-rich range, we evaluate all models using the standard test loss averaged over the full interval $t \in [0,1]$.

### 5.1. Locating the Signal-Rich Window

To validate our hypothesis regarding the inefficiency of extreme masking rates, we conducted an empirical ablation study using a 50M-parameter model adapted from the nanoGPT framework. The model was trained on the `WikiText` dataset (Merity et al., 2017) with a batch size of 1024 and a block size of 150 for 6000 steps. We partitioned the noise schedule $t \in [0,1]$ into ten intervals of width 0.1 (e.g., $t \in [0,0.1]$, $t \in [0.1,0.2]$, etc.). We then trained separate models where the masking ratio $t$ was sampled uniformly solely within each specific sub-interval. The results, illustrated in Figure 3, reveal a distinct U-shaped relationship between the masking interval and the final test loss. Please refer to Section F.3 for full loss vs time curves.

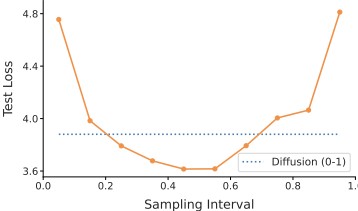

*Figure 3.* **Test Loss vs. Masking Interval.** Evaluation of 50M-parameter models trained on restricted masking ranges of width 0.1. The x-axis represents the midpoint of the sampling interval (e.g., $x = 0.05$ corresponds to $t \sim \mathcal{U}[0,0.1]$). The dashed blue line represents the baseline performance of standard full-range sampling ($t \sim \mathcal{U}[0,1]$). The U-shaped curve demonstrates that training exclusively within the "Signal-Rich" window ($t \in [0.4,0.5]$ or $[0.5,0.6]$) significantly outperforms the standard practice of sampling the full range. As a result, we use the range $t \in [0.45,0.55]$ for the following 8B experiments.

**Inefficiency at Extremes.** Models trained on intervals near the boundaries ($t \in [0,0.1]$ or $t \in [0.9,1.0]$) perform poorly, yielding test losses significantly worse than the baseline. This aligns with our theoretical insight: the signal would vanish if $t$ goes to the boundaries.

**Superiority of the Middle Range.** The dashed black line indicates the performance of the standard baseline ($t \sim \mathcal{U}[0,1]$), which yields a test loss of approximately 3.88. Crucially, models trained exclusively within the middle intervals ($t \in [0.4,0.5]$ or $[0.5,0.6]$) achieve losses as low as 3.62. This suggests that the standard practice of uniform sampling over $[0,1]$ is suboptimal because it allocates significant computational budget to low-signal regimes. By restricting the diffusion schedule to the window where the signal-to-noise ratio is most informative, we can achieve superior generalization. Section F.4 gives a theoretical derivation that predicts this optimum directly from corpus statistics.

### 5.2. Scalability to Large Language Models (8B)

#### 5.2.1. PRETRAINING

To further verify if the theoretical insights of the signal regime translate to larger scales, we utilize `dllm` framework (Zhou et al., 2026) to pretrain two versions of `LLaDA-8B` architecture (Nie et al., 2025b) from scratch on `DCLM-baseline` dataset (Li et al., 2024). Both models were trained with a batch size of 128 and block size of 4096 for 15,000 steps. We prioritize the evolution of evaluation loss as our primary metric, as it provides a stable signal of optimization efficiency during the early stages of pre-training. As illustrated in Figure 4, the model trained exclusively within the signal-rich window ($t \in [0.45,0.55]$, as suggested by Figure 3) exhibits a markedly faster reduction in perplexity compared to the standard $\mathcal{U}[0,1]$ baseline.

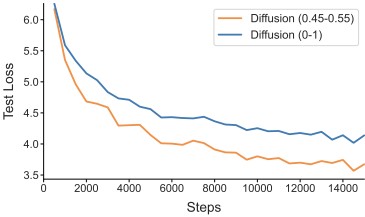

*Figure 4.* **Pre-training Test Loss (LLaDA-8B).** Evolution of the held-out negative log-likelihood (NLL) over 15K steps. The model trained with the restricted schedule $t \in [0.45, 0.55]$ (orange curve) consistently achieves lower loss compared to the standard $t \in [0, 1]$ baseline (blue curve), indicating superior training efficiency.

To complement the loss dynamics, we select `HellaSwag` (Zellers et al., 2019) and `ARC-Easy` (Clark et al., 2018) for downstream evaluation. These specific benchmarks are chosen because they are uniquely sensitive to the initial acquisition of commonsense and factual associations, whereas more complex reasoning tasks often remain at chance-level performance during early pre-training. For details of inference, we kindly refer readers to Section F.6.

Consistent with our theoretical findings on the $k$-parity task, the model trained exclusively within the signal-rich window ($t \in [0.45, 0.55]$) demonstrates a significantly faster acquisition of downstream capabilities. As shown in Table 1, the signal-optimized model achieves a 4.6% absolute improvement on `HellaSwag` and a substantial 8.8% gain on `ARC-Easy` over the uniform baseline at the same training step. These results empirically validate that strategic mask sampling, informed by our loss decomposition, effectively concentrates the optimization effort on the most informative regimes, and accelerating the reduction of the test loss and improving the overall training efficiency of the MD model.

*Table 1.* **Downstream Task Evaluation (Pre-training).** Comparison of zero-shot performance for LLaDA-8B models pretrained from scratch for 15,000 steps. We report accuracy on HellaSwag and ARC-Easy, as these benchmarks are sensitive to the emergence of basic linguistic features and common-sense reasoning during the early stages of pre-training. The signal-rich window ($t \in [0.45, 0.55]$) consistently outperforms the standard $\mathcal{U}[0, 1]$ baseline, reflecting higher training efficiency.

| Method | HellaSwag | ARC-Easy |
|---|---|---|
| PT ($t \in [0, 1]$) | 0.354 | 0.342 |
| **PT** ($t \in [0.45, 0.55]$) | **0.400** | **0.430** |

### 5.2.2. SUPERVISED FINE-TUNING

We further investigated whether these findings apply to fine-tuning. Starting from a pre-trained `LLaDA-8B Base` model, we performed SFT on the `tulu-3-sft-personas-math-filtered` dataset (Lambert et al., 2025) (See Section F.5 for details) for 1,200

steps (approx. 4 epochs) with a batch size of 256 and block size 1,024. For details of inference, we kindly refer readers to Section F.6.

**Evaluation on PPL-based Discriminative Tasks.** We first evaluate our models on multiple-choice benchmarks including `MMLU` (Hendrycks et al., 2021a), `ARC-Challenge`, and `GPQA` (Rein et al., 2024). As shown in Table 2, the results validate our "signal-rich" hypothesis. Our signal-rich window ($t \in [0.45, 0.55]$) demonstrates superior performance across all evaluated benchmarks compared to the standard $\mathcal{U}[0, 1]$ baseline. Most notably, in the context of knowledge-intensive reasoning (e.g., GPQA), our method achieves an accuracy of 0.402. This represents a substantial improvement over both the LLaDA Base model (0.252) and the vanilla SFT approach (0.344), which confirms that concentrating on the most informative levels enables the model to internalize complex patterns more efficiently.

*Table 2.* **Downstream Task Evaluation on PPL-based Discriminative Tasks (SFT).** All models were fine-tuned on a mathematical dataset. For multiple-choice benchmarks, the signal-rich window ($t \in [0.45, 0.55]$) consistently outperforms the uniform $t \in [0, 1]$ baseline, demonstrating better retention of general knowledge and more effective domain-specific learning.

| Method | MMLU | MMLU-stem | ARC-Challenge | GPQA |
|---|---|---|---|---|
| LLaDA Base | 0.659 | 0.629 | 0.459 | 0.252 |
| SFT ($t \in [0, 1]$) | 0.659 | 0.621 | 0.468 | 0.344 |
| **SFT** ($t \in [0.45, 0.55]$) | **0.669** | **0.635** | **0.480** | **0.402** |

**Evaluation on Generative Tasks.** Interestingly, we observe a performance divergence when moving to generative benchmarks. As illustrated in Table 3, the "signal-rich" window ($t \in [0.45, 0.55]$) that excelled in discriminative tasks actually underperforms the uniform $\mathcal{U}[0, 1]$ baseline on `GSM8K` (Cobbe et al., 2021) and `MATH` (Hendrycks et al., 2021b). We argue that for complex reasoning, while the $t \to 0$ regime remains uninformative, the near-total mask regime (where $t \to 1$) is not merely informative but essential. Intuitively, solving a difficult mathematical problem becomes significantly trivialized if even minimal "hints" (i.e., unmasked tokens) are provided. If a model is trained primarily on mid-range masking, it may learn to rely on these contextual anchors rather than developing the ability to "solve from scratch." To verify this, we conducted a range ablation by gradually shifting the sampling interval towards $t = 1$. As shown in Table 3, performance scales positively as we emphasize higher masking ratios, with the $[0.5, 1.0]$ configuration yielding the best results. This demonstrates that mastering logical reconstruction under extreme information scarcity is the key to superior generative performance, whereas the mid-range window is better suited for static knowledge recognition.

*Table 3.* **Generative Task Performance vs. Masking Range.** Evaluation of SFT models on `GSM8K` and `MATH`. Results show a positive correlation between performance and the emphasis on high-noise intervals. Shifting the sampling range toward $t = 1$ (e.g., $[0.5, 1.0]$) significantly outperforms the mid-range "signal-rich" window, demonstrating that mastering structural reconstruction is essential for complex generative reasoning.

| Task | Base | $[0.45, 0.55]$ | $[0, 1]$ | $[0.2, 1]$ | $[0.3, 1]$ | $[0.5, 1]$ |
|------|------|------|------|------|------|------|
| **GSM8K** | 0.703 | 0.738 | 0.768 | 0.762 | 0.774 | **0.785** |
| **MATH** | 0.314 | 0.342 | 0.341 | 0.358 | 0.360 | **0.375** |

## 6. Conclusion and Limitation

In this work, we characterized the structural properties of the MD objective, identifying a fundamental decomposition into a Signal Regime and an information-theoretically obscured Noise Regime. We proved that the latter functions as a powerful implicit regularizer, enabling models to bypass the grokking plateau and achieve rapid generalization on complex discrete structures like the $k$-parity problem. Furthermore, we demonstrated the scalability of these insights, showing that our strategy significantly improves performance at both the 50M and 8B parameter scales.

Despite these gains, this study has limitations that warrant future exploration. Our exploration was primarily limited to restricted uniform intervals of width $0.1$ while complex non-linear or curriculum-based schedules may yield further improvements. We hope this work encourages further research into the training dynamics of discrete diffusion models.

## Acknowledgements

This research was supported in part by the NSF CAREER Award 2146492, NSF-Simons AI Institute for Cosmic Origins (CosmicAI), and NSF AI Institute for Foundations of Machine Learning (IFML). JH thanks Zixuan Wang for sharing the blog post that became the starting point of this work, and Yunwei Ren for insightful discussions.

## Impact Statement

This paper presents work whose goal is to advance the field of Machine Learning. There are many potential societal consequences of our work, none which we feel must be specifically highlighted here.

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

# A. More related works of MDLM

### A.1. Training and Fine-tuning

Recent improvements typically focus on the spatial or token-wise importance within a sequence. Kosmopoulou et al. (2025) assigns higher masking probabilities or loss weights to rare tokens. The intuition is to prioritize learning from infrequent but semantically rich tokens, preventing the model from being biased toward common function words. Dream (Ye et al., 2025) proposes reweighting each masked position based on the intuition that clean tokens in the neighborhood should exert a greater influence on the prediction of masked tokens, thereby improving context comprehension. Other works propose mask-agnostic objectives to enforce prediction invariance to mask counts (Piskorz et al., 2025) or utilize Reinforcement Learning (RL) for fine-tuning (Zhao et al., 2025b;c; Wang et al., 2025a; He et al., 2025; Gong et al., 2025b; Zhao et al., 2025a).

A closely related direction reweights the objective along the temporal (noise-level) dimension rather than the spatial one. Shi & Titsias (2025) interpret reweighted losses as a cascade of time-dependent variational bounds that provably tighten the ELBO, yielding a reweighting scheme that applies to both continuous Gaussian and discrete (masked) diffusion. Our analysis offers a distinct and *complementary* perspective to this loss-centric view. Whereas Shi & Titsias (2025) justify reweighting by constructing tighter bounds, such bounds do not capture the actual optimization dynamics, which differ fundamentally between modalities. This distinction is consequential: as illustrated by the grokking phenomenon in Figure 2, attaining an exact (or better) loss does not necessarily translate into a better learning process. Our signal-to-noise analysis supplies this missing link, showing that adjusting the masking ratio $t$—which itself corresponds to a reweighting schedule—inherently acts as a regularizer (Theorem 4.3) and controls the effective learning rate (Theorem 4.5). These optimization-level effects are not recoverable from loss bounds such as Shi & Titsias (2025) alone, and the resulting theoretically derived $t$ improves not only final performance but also convergence and training efficiency (Figure 2).

### A.2. Sampling during Inference

Another line of research focuses on improving the generation process at test-time. This includes optimizing unmasking(Hong et al., 2025a; Jazbec et al., 2025; Wu et al., 2025; Huang et al., 2026; Wang et al., 2025a; Kim et al., 2025; Azangulov et al., 2025) and remasking (Hong et al., 2025b; Dong et al., 2025; He et al., 2025) strategies to find better decoding trajectories. Since these are test-time optimizations aimed at improving generation quality, they do not address the training-time signal-to-noise dynamics analyzed in our work.

### A.3. Hybrid Architectures

Recent works explore the combination of ARMs and DLMs. For instance, TiDAR (Liu et al., 2025) introduces a "Thinking-Talking" paradigm, utilizing diffusion for parallel drafting and AR-logic for sampling within a single forward pass via structured attention masks. Similarly, Block Diffusion (Arriola et al., 2025) and D2F (Wang et al., 2025b) interpolates between the two by performing discrete diffusion at a block level, supporting flexible-length generation and KV caching. While these methods innovate on the sequence-level structure and drafting efficiency, their diffusion components still utilize standard training objectives and timestep schedules.

# B. Proof of Theorem 4.2 and Corollary 4.7

### B.1. Proof of Theorem 4.2

To derive the sample complexity bound, we first formalize the notion of the theoretical minimum loss (Bayes Risk) for a specific feature $i$ under the stochastic masking process.

**Definition B.1** (Bayes Optimal Risk)**.** For a target bit $x_i'$ and a masking rate $t$, let $\tilde{\boldsymbol{x}}$ denote the corrupted input vector. The Bayes optimal risk $R_i^*(t)$ is defined as the minimum expected cross-entropy loss achieving by any function $f$:

$$R_i^*(t) = \inf_f \mathbb{E}_{\mathbf{m} \sim t} \left[ \mathrm{CE}(x_i', f(\tilde{\boldsymbol{x}})) \mid i \in M_{\mathbf{m}} \right] = \mathbb{E}_{\mathbf{m} \sim t} \left[ H(x_i' \mid \tilde{\boldsymbol{x}}_{\backslash i}) \right].$$

where $H(x_i' \mid \tilde{\boldsymbol{x}}_{\backslash i})$ is the conditional entropy of the masked bit given the visible tokens.

We can explicitly calculate $R_i^*(t)$ for noise and signal features.

1. Noise Features ($i \notin \mathcal{S}'$).

   Since the input bits are drawn uniformly $x_j \sim \text{Unif}\{-1, 1\}$, a noise bit is statistically independent of all other bits. The posterior distribution is always uniform: $P(x_i' = 1|\tilde{x}) = 0.5$. The entropy is maximal:

   $$R_i^*(t) = -\left(\frac{1}{2}\log\frac{1}{2} + \frac{1}{2}\log\frac{1}{2}\right) = \log 2.$$

2. Parity Features ($i \in \mathcal{S}'$).

   The parity bit is deterministic given the other $k$ bits in the secret set $\mathcal{S}'$. In the identifiable case where all other $k$ bits in $\mathcal{S}' \setminus \{i\}$ are unmasked (which happens with probability $(1-t)^k$), the conditional entropy is 0. Otherwise, if any of the other $k$ bits are masked, the partial view gives no information about the parity. The posterior reverts to uniform, and the entropy is $\log 2$.

Thus, the expected risk is:

$$R_i^*(t) = (1-t)^k \cdot 0 + (1 - (1-t)^k) \cdot \log 2 = (1 - (1-t)^k)\log 2.$$

With these definitions established, we proceed to the proof.

We restate the following theorem.

**Theorem B.2** (Sample Complexity for Support Identification). *Let $\mathcal{D}'$ be a dataset of $N$ samples subjected to the stochastic masking process in Definition 3.4. To ensure that the secret set $S$ is uniquely identifiable by the marginal loss gap with probability at least $1 - \delta$, the number of samples $N$ must satisfy:*

$$N \geq \frac{4\log(4n/\delta)}{\mathbb{E}[t] \cdot \left(\mathbb{E}[(1-t)^k]\right)^2}.$$

*where $k = |\mathcal{S}|$ is the size of the secret set (excluding the parity bit itself).*

*Proof.* The goal is to identify the secret set $\mathcal{S}$ by exploiting the difference in expected loss between parity features and noise features. For a feature $i$, we define the theoretical minimum reconstruction loss under masking as $R_i^*(t)$.

If $i \notin \mathcal{S}'$, the bit is independent of all other bits; hence, when masked, it is completely unpredictable:

$$R_i^*(t) = \log 2.$$

If $i \in \mathcal{S}'$, the bit is determined by the other $k$ bits in $\mathcal{S}'$. It is recoverable if and only if all other $k$ bits are unmasked (which occurs with probability $(1-t)^k$). Otherwise, it is unpredictable:

$$R_i^*(t) = (1 - (1-t)^k)\log 2.$$

Averaging over $t$, we define the expected loss $\bar{R}_i^* = \mathbb{E}_t[R_i^*(t)]$. The gap between noise and signal features is:

$$\Delta = \bar{R}_{i\notin S}^* - \bar{R}_{i\in S}^* = \log 2 - (1 - \rho_k)\log 2 = \rho_k \log 2,$$

where $\rho_k = \mathbb{E}_t[(1-t)^k]$.

Let $m_i$ be the number of times feature $i$ is masked in the dataset. Let $\hat{R}_i$ be the empirical cross-entropy loss computed on these $m_i$ samples. By Hoeffding's inequality, the probability that the empirical loss deviates from the true mean by more than a tolerance $\epsilon$ is:

$$\Pr\left(\left|\hat{R}_i - \bar{R}_i^*\right| \geq \epsilon\right) \leq 2\exp\left(-\frac{2m_i\epsilon^2}{(\log 2)^2}\right).$$

To reliably distinguish signal from noise, we set the tolerance $\epsilon = \Delta/2$. If this condition holds, the error intervals for signal and noise features will not overlap.

We require this condition to hold simultaneously for all $n$ features with probability at least $1 - \delta/2$. Applying a union bound:

$$\Pr\left(\exists i, \left|\hat{R}_i - \bar{R}_i^*\right| \geq \frac{\Delta}{2}\right) \leq 2n \exp\left(-\frac{2m_{\min}(\Delta/2)^2}{(\log 2)^2}\right) \leq \frac{\delta}{2}.$$

Substituting $\Delta = \rho_k \log 2$ and solving for the minimum required mask count $m_{\min}$:

$$2n \exp\left(-\frac{1}{2}m_{\min}\rho_k^2\right) \leq \frac{\delta}{2}$$

$$-\frac{1}{2}m_{\min}\rho_k^2 \leq \ln\left(\frac{\delta}{4n}\right) = -\ln\left(\frac{4n}{\delta}\right)$$

$$m_{\min} \geq \frac{2}{\rho_k^2}\ln\left(\frac{4n}{\delta}\right).$$

Finally, we ensure that the dataset size $N$ is large enough such that every feature is masked at least $m_{\min}$ times. The number of masks $m_i$ for any bit follows a binomial distribution $m_i \sim \text{Bin}(N, \mathbb{E}[t])$.

We apply the multiplicative Chernoff bound to bound the probability that $m_i$ falls below half its expectation, i.e., $m_i \leq \frac{1}{2}N\mathbb{E}[t]$:

$$\Pr\left(m_i \leq \frac{1}{2}N\mathbb{E}[t]\right) \leq \exp\left(-\frac{1}{8}N\mathbb{E}[t]\right).$$

Applying a union bound over all $n$ features, we require this failure probability to be at most $\delta/2$:

$$n \exp\left(-\frac{1}{8}N\mathbb{E}[t]\right) \leq \frac{\delta}{2} \implies N \geq \frac{8}{\mathbb{E}[t]}\ln\left(\frac{2n}{\delta}\right).$$

Combining the requirements, we have

$$\frac{1}{2}N\mathbb{E}[t] \geq \frac{2}{\rho_k^2}\ln\left(\frac{4n}{\delta}\right).$$

Solving for $N$ yields the final bound:

$$N \geq \frac{4\ln(4n/\delta)}{\mathbb{E}[t] \cdot \rho_k^2}.$$

This completes the proof.

$\square$

## B.2. Proof of Corollary 4.7

We restate the corollary as follows.

**Corollary B.3** (Sample Complexity-Optimal Masking Rate). *Assume the masking rate follows a uniform distribution $t \sim U[t_0, t_1]$. The sample complexity objective dictates the optimal strategy based on the dependency order $k$:*

1. **Simple Dependencies** ($k = 1$): *Any uniform distribution is optimal provided the mean is:*

$$\mathbb{E}[t] = \frac{1}{3}.$$

   *This allows for any range $[t_0, t_1]$ satisfying $t_0 + t_1 = 2/3$ (e.g., fixed $t = 1/3$ or $U[0, 2/3]$).*

2. **Complex Dependencies** ($k > 1$): *The optimal distribution is anchored at the lower boundary $t_0 = 0$. The upper bound $t_1 \in (0, 1)$ is the root of:*

$$(2k + 1)(1 - t_1)^{k+1} - (2k + 2)(1 - t_1)^k + 1 = 0.$$

*Proof.* We analyze the sample complexity objective $J$ for a uniform masking distribution $t \sim U[t_0, t_1]$. The objective is given by:

$$J(t_0, t_1) = \mathbb{E}[t] \cdot \left( \mathbb{E}[(1 - t)^k] \right)^2. \tag{4}$$

For the uniform distribution, the moments are $\mathbb{E}[t] = \frac{t_0 + t_1}{2}$ and $\mathbb{E}[(1-t)^k] = \frac{(1-t_0)^{k+1} - (1-t_1)^{k+1}}{(t_1 - t_0)(k+1)}$. Dropping constant factors which do not affect the location of the optimum, we maximize

$$\hat{J}(t_0, t_1) = (t_0 + t_1) \left[ \frac{(1 - t_0)^{k+1} - (1 - t_1)^{k+1}}{t_1 - t_0} \right]^2.$$

Let $x = 1 - t_0$ and $y = 1 - t_1$. The constraints $0 \le t_0 \le t_1 \le 1$ map to $1 \ge x \ge y \ge 0$. The transformed objective function $G(x, y)$ is

$$G(x, y) = (2 - (x + y)) \left( \frac{x^{k+1} - y^{k+1}}{x - y} \right)^2.$$

Let $y = rx$, where $r \in [0, 1]$. We substitute this into the term inside the square

$$\frac{x^{k+1} - (rx)^{k+1}}{x - rx} = x^k \frac{1 - r^{k+1}}{1 - r}.$$

Substituting this back into $G(x, y)$, we isolate the dependency on $x$ and $r$

$$G(x, r) = \left( \frac{1 - r^{k+1}}{1 - r} \right)^2 \cdot x^{2k} (2 - x(1 + r)).$$

We denote $A(r) = \left( \frac{1 - r^{k+1}}{1 - r} \right)^2$ and $\Phi(x) = x^{2k}(2 - x(1 + r))$. We first maximize $\Phi(x)$ for a fixed ratio $r$. The derivative is

$$\begin{aligned}
\Phi'(x) &= 2k x^{2k-1}(2 - x(1 + r)) + x^{2k}(-(1 + r)) \\
&= x^{2k-1} \left[ 4k - 2kx(1 + r) - x(1 + r) \right] \\
&= x^{2k-1} \left[ 4k - (1 + r)(2k + 1)x \right].
\end{aligned}$$

Setting $\Phi'(x) = 0$ yields the unique stationary point:

$$x^*(r) = \frac{4k}{(2k + 1)(1 + r)}.$$

The function $\Phi(x)$ strictly increases for $x < x^*(r)$ and decreases for $x > x^*(r)$. This defines two regimes:

- $x^*(r) \le 1$: This occurs when $r \ge \frac{2k-1}{2k+1}$. Here, the peak lies within the valid range, so the optimal scale is $x = x^*(r)$.

- $x^*(r) > 1$: This occurs when $r < \frac{2k-1}{2k+1}$. Here, the unconstrained peak lies to the right of the feasible region ($x > 1$). Since $\Phi(x)$ is strictly increasing on the feasible interval $[0, 1]$, the maximum is constrained to the boundary. Thus, the optimal scale is clamped at $x = 1$.

Next, we only consider the case where $x^*(r) \le 1$ and $r \ge \frac{2k-1}{2k+1}$. We substitute the optimal $x^*(r)$ back into the objective function to analyze the dependency on the distribution shape $r$. Since $2 - x^*(r)(1 + r) = 2 - \frac{4k}{2k+1} = \frac{2}{2k+1}$, the value function $G(r)$ scales as:

$$G(r) \propto A(r) \cdot (x^*(r))^{2k} \propto \left( \frac{1 - r^{k+1}}{1 - r} \right)^2 \left( \frac{1}{1 + r} \right)^{2k}.$$

When $k = 1$, $G(r)$ is constant w.r.t $r$. Directly using Equation (4), we can get the only constraint is $\mathbb{E}[t] = \frac{1}{3}$.

When $k > 1$, we have

$$G'(r) = \frac{\mathrm{d}\left(\left(\frac{1-r^{k+1}}{1-r}\right)^2 \left(\frac{1}{1+r}\right)^{2k}\right)}{\mathrm{d}r} = -\frac{2\left(\frac{1}{r+1}\right)^{2k+1}\left(r^{k+1}-1\right)\left(k(r-1)\left(r^k+1\right)-(r+1)\left(r^k-1\right)\right)}{(1-r)^3}.$$

Since $1 + r^k \geq r^i + r^{k-i}$ for any $1 \leq i \leq k-1$ and $0 \leq r \leq 1$, we have

$$(k-1) + (k-1)r^k \geq 2\left(r + r^2 + \cdots + r^{k-1}\right),$$
$$k + kr^k \geq 1 + 2\left(r + r^2 + \cdots + r^{k-1}\right) + r^k,$$
$$k\left(r^k + 1\right) \geq (r+1)\left(1 + r + r^2 + \cdots + r^{k-1}\right),$$
$$k\left(r^k + 1\right)(r-1) \leq (r+1)\left(r^k - 1\right).$$

Therefore, $G'(r) \leq 0$ and $G(r)$ is monotonically non-increasing w.r.t $\frac{2k-1}{2k+1} \leq r \leq 1$. Thus, the optimal would be at the point where $r^* = \frac{2k-1}{2k+1}$ and $x^* = 1$.

Since $x^* = 1$ at any case when $k > 1$, we maximize the objective with respect to $y$ along the boundary

$$G(1, y) = (1-y)\left(\frac{1-y^{k+1}}{1-y}\right)^2 = \frac{(1-y^{k+1})^2}{1-y}.$$

Differentiating $\ln G(1, y)$ with respect to $y$ and setting to zero

$$\frac{-2(k+1)y^k}{1-y^{k+1}} + \frac{1}{1-y} = 0.$$

Rearranging terms leads to the defining polynomial for the optimal upper bound

$$(2k+1)y^{k+1} - (2k+2)y^k + 1 = 0.$$

$\square$

## C. Proof of Theorem 4.3 and Remark 4.4

### C.1. Proof of Theorem 4.3

We first restate the theorem.

**Theorem C.1** (Effective Loss Decomposition). *Consider the training objective Equation* (1) *over a finite dataset $\mathcal{D}'$. For $\boldsymbol{m} \in \mathcal{R}_S$, we define $f^*(\tilde{\boldsymbol{z}}) = \frac{1}{|M_{\boldsymbol{m}}|}x'_{M_{\boldsymbol{m}} \cap S'}$ to be the optimal target function. The loss is equivalent to:*

$$\mathcal{L}_{\mathrm{eff}}(\theta) = P_S \mathbb{E}_{\tilde{\boldsymbol{z}}|\mathcal{R}_S}\left[\frac{|M_{\boldsymbol{m}}|}{2t}(f_\theta(\tilde{\boldsymbol{z}}) - f^*(\tilde{\boldsymbol{z}}))^2\right] + P_N \mathbb{E}_{\tilde{\boldsymbol{z}}|\mathcal{R}_N}\left[\frac{|M_{\boldsymbol{m}}|}{2t}f_\theta(\tilde{\boldsymbol{z}})^2\right]$$

*where $P_S = (k+1)\mathbb{E}_{t \sim U[t_0, t_1]}\left[t(1-t)^k\right]$ and $P_N = 1 - P_S$. The expectations are taken over the empirical distribution of $\tilde{\boldsymbol{z}}$ conditioned on the respective regimes.*

*Proof.* We recall that the overall training objective is given by

$$\mathcal{L}(\theta) = \mathbb{E}_{\boldsymbol{x}' \in \mathcal{D}', t \sim U[t_0, t_1], \boldsymbol{m}}\left[\frac{1}{2t}\sum_{j \in M_{\boldsymbol{m}}}\left(\mathrm{TF}_\theta(\widetilde{\boldsymbol{X}})_{[:,j]} - x'_j\right)^2\right]$$

$$= \mathbb{E}_{\tilde{\boldsymbol{z}}}\left[\frac{1}{2t}\sum_{j \in M_{\boldsymbol{m}}}\left(f_\theta(\tilde{\boldsymbol{z}}) - x'_j\right)^2\right]$$

$$= \mathbb{E}_{\tilde{\boldsymbol{z}}}\left[\frac{1}{2t}\left(|M_{\boldsymbol{m}}|f_\theta(\tilde{\boldsymbol{z}})^2 - 2\left(\sum_{j \in M_{\boldsymbol{m}}}x'_j\right)f_\theta(\tilde{\boldsymbol{z}}) + \frac{\left(\sum_{j \in M_{\boldsymbol{m}}}x'_j\right)^2}{|M_{\boldsymbol{m}}|}\right)\right] + \mathbb{E}_{\tilde{\boldsymbol{z}}}\left[\frac{1}{2t}\left(\sum_{j \in M_{\boldsymbol{m}}}x_j'^2 - \frac{\left(\sum_{j \in M_{\boldsymbol{m}}}x'_j\right)^2}{|M_{\boldsymbol{m}}|}\right)\right].$$

We denote $C_0 = \mathbb{E}_{\tilde{z}}\left[\frac{1}{2t}\left(\sum_{j\in M_m} x_j'^2 - \frac{(\sum_{j\in M_m} x_j')^2}{|M_m|}\right)\right]$, then the training objective reduces to

$$\mathcal{L}(\theta) = \mathbb{E}_{x',t,m}\left[\frac{|M_m|}{2t}\left(f_\theta(\tilde{z}) - \frac{\sum_{j\in M_m} x_j'}{|M_m|}\right)^2\right] + C_0$$

$$= \mathbb{E}_{t,m}\left[\frac{|M_m|}{2t}\mathbb{E}_{x'|t,m}\left[\left(f_\theta(\tilde{z}) - \frac{\sum_{j\in M_m} x_j'}{|M_m|}\right)^2\right]\right] + C_0.$$

If $m \in \mathcal{R}_N$, we consider the following two cases. If $|M_m \cap S'| = 0$, then the masked positions are irrelevant to the parity task. Therefore $f_\theta(\tilde{z})$ is independent to $\sum_{j\in M_m} x_j'$. In the second case, if $|M_m \cap S'| \geq 2$, then for $j \in M_m \cap S'$, $x_j'$ is orthogonal to any function of $\tilde{z}$. In both cases, we have

$$\mathbb{E}_{x'|t,m}\left[f_\theta(\tilde{z})\sum_{j\in M_m} x_j'\right] = 0.$$

If $m \in \mathcal{R}_S$, then we denote $M_m \cap S' = \{k\}$. For $j \in M_m \backslash \{k\}$, $f_\theta(\tilde{z})$ is independent to $x_j'$. Therefore, we have

$$\mathbb{E}_{x'|t,m}\left[f_\theta(\tilde{z})\sum_{j\in M_m} x_j'\right] = \mathbb{E}_{x'|t,m}[f_\theta(\tilde{z})x_k'].$$

Therefore, absorb all constant to $C_1$, we can then reduce the loss to

$$\mathcal{L}(\theta) = \Pr(m \in \mathcal{R}_S) \cdot \mathbb{E}_{\tilde{z}|\mathcal{R}_S}\left[\frac{|M_m|}{2t}\left(f_\theta(\tilde{z}) - \frac{x_k'}{|M_m|}\right)^2\right] + \Pr(m \in \mathcal{R}_N) \cdot \mathbb{E}_{\tilde{z}|\mathcal{R}_N}\left[\frac{|M_m|}{2t}f_\theta(\tilde{z})^2\right] + C_1.$$

We thus finish the proof by calculating

$$\Pr(m \in \mathcal{R}_S) = (k+1)\mathbb{E}_{t\sim U[t_0,t_1]}\left[t(1-t)^k\right].$$

$\square$

## C.2. Proof of Remark 4.4

For convenience, we first restate the result and then give its full derivation. With a slight abuse of notation, we overload $f_\theta$ as $f_\theta(\tilde{z}) = \text{sigmoid}\left(v^\top \sigma(W\tilde{z})\right)$.

**Theorem C.2** (Extension to the Cross-Entropy Loss; restatement of Remark 4.4). *Let $f_\theta(\tilde{z}) \in (0,1)$ denote the predicted probability and define the regime-dependent targets*

$$f_S^*(\tilde{z}) = \frac{x_k' + (|M_m| - 1)/2}{|M_m|} \text{ for } m \in \mathcal{R}_S, M_m \cap S' = \{k\}, \qquad f_N^*(\tilde{z}) = \tfrac{1}{2} \text{ for } m \in \mathcal{R}_N.$$

*Then the binary cross-entropy objective is equivalent to*

$$\mathcal{L}_{\text{eff}}(\theta) = \sum_{R\in\{\mathcal{R}_S,\mathcal{R}_N\}} \Pr(m \in R)\, \mathbb{E}_{\tilde{z}|R}\left[\frac{|M_m|}{2t} D_{\text{KL}}(f_R^*(\tilde{z}) \,\|\, f_\theta(\tilde{z}))\right].$$

*Proof.* By the law of total expectation, we rewrite the objective by conditioning on the corrupted state $\tilde{z}$:

$$\mathcal{L}(\theta) = \mathbb{E}_{t,m}\left[\frac{1}{2t}\mathbb{E}_{z|t,m}\left[\sum_{j\in M_m}\mathbb{E}_{x'|\tilde{z}}\left[\text{BCE}(x_j', f_\theta(\tilde{z}))\right]\right]\right]. \tag{5}$$

Using the identity $\text{BCE}(p,q) = D_{\text{KL}}(p \,\|\, q) + H(p)$, we evaluate the inner sum in each regime; note that $\text{BCE}(p,q)$ is linear in its first argument $p$.

**Case 1: $\boldsymbol{m} \in \mathcal{R}_N$.** We have $\mathbb{E}[x'_j \mid \tilde{z}] = \frac{1}{2}$ for every $j \in M_{\boldsymbol{m}}$. The inner sum becomes

$$|M_{\boldsymbol{m}}| \operatorname{BCE}(\tfrac{1}{2}, f_\theta) = |M_{\boldsymbol{m}}| D_{\mathrm{KL}}(\tfrac{1}{2} \| f_\theta) + C_0, \tag{6}$$

where $C_0 = |M_{\boldsymbol{m}}| H(\frac{1}{2})$ is independent of $\theta$.

**Case 2: $\boldsymbol{m} \in \mathcal{R}_S$.** Here $M_{\boldsymbol{m}} \cap \mathcal{S}' = \{k\}$, while the remaining $|M_{\boldsymbol{m}}| - 1$ masked tokens are noise with $\mathbb{E}[x'_j \mid \tilde{z}] = \frac{1}{2}$. By linearity of BCE in its first argument, averaging the targets gives

$$\operatorname{BCE}(x'_k, f_\theta) + (|M_{\boldsymbol{m}}| - 1) \operatorname{BCE}(\tfrac{1}{2}, f_\theta) = |M_{\boldsymbol{m}}| \operatorname{BCE}(f_S^*(\tilde{z}), f_\theta) = |M_{\boldsymbol{m}}| D_{\mathrm{KL}}(f_S^*(\tilde{z}) \| f_\theta) + C_1, \tag{7}$$

with $C_1 = |M_{\boldsymbol{m}}| H(f_S^*(\tilde{z}))$ independent of $\theta$.

Substituting both cases back into the outer expectations and splitting by the regime probabilities $\Pr(\boldsymbol{m} \in \mathcal{R}_S)$ and $\Pr(\boldsymbol{m} \in \mathcal{R}_N)$ yields the stated identity, completing the proof. $\qquad \square$

## D. Proof of Theorem 4.5 and Corollary 4.6

### D.1. Proof of Theorem 4.5

**Theorem D.1** (Masked Diffusion Energy Landscape). *Consider the effective loss $\mathcal{L}_{\mathrm{eff}}(\boldsymbol{v}, \boldsymbol{W})$ defined in Equation (3). Let $\boldsymbol{h}(\tilde{z}) = \sigma(\boldsymbol{W}\tilde{z})$. Let the correlation vector $\boldsymbol{c}(\boldsymbol{W})$ and the covariance matrix $\boldsymbol{\Sigma}(\boldsymbol{W})$ be defined as*

$$\boldsymbol{c}(\boldsymbol{W}) = P_S \mathbb{E}_S \left[ \frac{|M_{\mathbf{m}}|}{2t} f^* \boldsymbol{h} \right]$$

$$\boldsymbol{\Sigma}(\boldsymbol{W}) = P_S \mathbb{E}_S \left[ \frac{|M_{\mathbf{m}}|}{2t} \boldsymbol{h}\boldsymbol{h}^\top \right] + P_N \mathbb{E}_N \left[ \frac{|M_{\mathbf{m}}|}{2t} \boldsymbol{h}\boldsymbol{h}^\top \right].$$

*We assume the readout $\boldsymbol{v}$ is at optimality for a fixed $\boldsymbol{W}$. Then minimizing the effective loss is equivalent to maximizing the energy $E(\boldsymbol{W})$*

$$E(\boldsymbol{W}) = \boldsymbol{c}(\boldsymbol{W})^\top \boldsymbol{\Sigma}(\boldsymbol{W})^\dagger \boldsymbol{c}(\boldsymbol{W}).$$

*Proof.* Recall the effective loss in Equation (3)

$$\mathcal{L}_{\mathrm{eff}}(\theta) = P_S \mathbb{E}_{\tilde{z}|\mathcal{R}_S} \left[ \frac{|M_{\boldsymbol{m}}|}{2t} (f_\theta(\tilde{z}) - f^*(\tilde{z}))^2 \right] + P_N \mathbb{E}_{\tilde{z}|\mathcal{R}_N} \left[ \frac{|M_{\boldsymbol{m}}|}{2t} f_\theta(\tilde{z})^2 \right],$$

$$= \boldsymbol{v}^\top \left( P_S \mathbb{E}_S \left[ \frac{|M_{\boldsymbol{m}}|}{2t} \boldsymbol{h}\boldsymbol{h}^\top \right] + P_N \mathbb{E}_N \left[ \frac{|M_{\boldsymbol{m}}|}{2t} \boldsymbol{h}\boldsymbol{h}^\top \right] \right) \boldsymbol{v} - 2\boldsymbol{v}^\top \left( P_S \mathbb{E}_S \left[ \frac{|M_{\boldsymbol{m}}|}{2t} f^* \boldsymbol{h} \right] \right) + P_S \mathbb{E}_S \left[ \frac{|M_{\boldsymbol{m}}|}{2t} f^{*2} \right]$$

We denote

$$\boldsymbol{\Sigma}(\boldsymbol{W}) = P_S \mathbb{E}_S \left[ \frac{|M_{\boldsymbol{m}}|}{2t} \boldsymbol{h}\boldsymbol{h}^\top \right] + P_N \mathbb{E}_N \left[ \frac{|M_{\boldsymbol{m}}|}{2t} \boldsymbol{h}\boldsymbol{h}^\top \right],$$

$$\boldsymbol{c}(\boldsymbol{W}) = P_S \mathbb{E}_S \left[ \frac{|M_{\boldsymbol{m}}|}{2t} f^* \boldsymbol{h} \right].$$

For any fixed $\boldsymbol{W}$, the optimal readout $\boldsymbol{v}^*$ is not unique if $\boldsymbol{\Sigma}(\boldsymbol{W})$ is rank-deficient. However, the set of minimizers forms an affine space, and every point in this set yields the identical effective loss $\mathcal{L}_{\mathrm{eff}}$. For simplicity, we choose

$$\boldsymbol{v}^* = \boldsymbol{\Sigma}(\boldsymbol{W})^\dagger \boldsymbol{c}(\boldsymbol{W}),$$

which is the unique minimum-norm solution. Substituting $\boldsymbol{v}^*$ back into $\mathcal{L}_{\mathrm{eff}}$, we have

$$\mathcal{L}_{\mathrm{eff}}(\boldsymbol{v}^*, \boldsymbol{W}) = \mathrm{Const} - \boldsymbol{c}^\top \boldsymbol{\Sigma}^\dagger \boldsymbol{c} = \mathrm{Const} - E(\boldsymbol{W}).$$

This proves that minimizing the effective loss is equivalent to maximizing $E(\boldsymbol{W})$. $\qquad \square$

### D.2. Proof of Corollary 4.6

**Corollary D.2** (Collapse of Feature Learning in Pure Signal Regime). *Consider the case where the training falls entirely into the Signal Regime ($P_N \to 0$). If the network is over-parameterized such that the target vector $\boldsymbol{y} \in \mathbb{R}^T$ lies within the subspace spanned by the feature activations $\boldsymbol{H} \in \mathbb{R}^{D \times T}$, the energy function degenerates to a data-dependent constant:*

$$E(\boldsymbol{W}) \xrightarrow{P_N \to 0} \frac{1}{T}\|\boldsymbol{y}\|^2 = \text{Constant}.$$

*In this regime, $\nabla_{\boldsymbol{W}} E(\boldsymbol{W}) = \boldsymbol{0}$, implying no gradient pressure to learn structured representations.*

*Proof.* When $P_N = 0$, the expectations are defined over the Signal Regime. Let $T$ be the number of samples. We concatenate the feature vectors $\boldsymbol{h}(\tilde{\boldsymbol{z}}_i)$ into a matrix $\boldsymbol{H} \in \mathbb{R}^{D \times T}$ and the target values $f^*(\tilde{\boldsymbol{z}}_i)$ into a vector $\boldsymbol{y} \in \mathbb{R}^T$:

$$\boldsymbol{H} = \left[ \sqrt{\frac{|M_{\boldsymbol{m}_1}|}{2t_1}}\boldsymbol{h}(\tilde{\boldsymbol{z}}_1) \quad \cdots \quad \sqrt{\frac{|M_{\boldsymbol{m}_T}|}{2t_T}}\boldsymbol{h}(\tilde{\boldsymbol{z}}_T) \right], \quad \boldsymbol{y} = \begin{bmatrix} \sqrt{\frac{|M_{\boldsymbol{m}_1}|}{2t_1}}f^*(\tilde{\boldsymbol{z}}_1) \\ \vdots \\ \sqrt{\frac{|M_{\boldsymbol{m}_T}|}{2t_T}}f^*(\tilde{\boldsymbol{z}}_T) \end{bmatrix}.$$

The empirical covariance and correlation can be written in matrix form as

$$\Sigma_S = \frac{1}{T}\boldsymbol{H}\boldsymbol{H}^\top, \quad \boldsymbol{c} = \frac{1}{T}\boldsymbol{H}\boldsymbol{y}.$$

Substituting these into the energy equation $E(\boldsymbol{W}) = \boldsymbol{c}^\top \Sigma_S^\dagger \boldsymbol{c}$:

$$E(\boldsymbol{W}) = \left(\frac{1}{T}\boldsymbol{H}\boldsymbol{y}\right)^\top \left(\frac{1}{T}\boldsymbol{H}\boldsymbol{H}^\top\right)^\dagger \left(\frac{1}{T}\boldsymbol{H}\boldsymbol{y}\right)$$
$$= \frac{1}{T}\boldsymbol{y}^\top \boldsymbol{H}^\top (\boldsymbol{H}\boldsymbol{H}^\top)^\dagger \boldsymbol{H}\boldsymbol{y}.$$

We recognize the term $\boldsymbol{P} = \boldsymbol{H}^\top(\boldsymbol{H}\boldsymbol{H}^\top)^\dagger\boldsymbol{H}$ as the orthogonal projection matrix onto the row space of $\boldsymbol{H}$. Under the over-parameterization assumption, the network is expressive enough to represent the target perfectly, meaning $\boldsymbol{y}$ lies in the row space of $\boldsymbol{H}$. Therefore, the projection preserves the vector: $\boldsymbol{P}\boldsymbol{y} = \boldsymbol{y}$. The energy simplifies to:

$$E(\boldsymbol{W}) = \frac{1}{T}\boldsymbol{y}^\top\boldsymbol{y} = \frac{1}{T}\sum_{i=1}^{T}\frac{|M_{\boldsymbol{m}_i}|}{2t_i}f^*(\tilde{\boldsymbol{z}}_i)^2.$$

This value depends solely on the dataset labels and is independent of the weights $\boldsymbol{W}$. Consequently, the energy landscape is flat, and feature learning collapses. $\qquad\square$

## E. Proof of Corollary 4.8

**Corollary E.1** (Signal-Optimal Masking Rate). *Assume the masking rate follows a uniform distribution $t \sim U[t_0, t_1]$. The signal-noise ratio dictates the optimal strategy by maximizing $P_S$ as*

$$t_0 = t_1 = \frac{1}{k+1}.$$

*Proof.* By Theorem 4.3 and Theorem 4.5, we get the signal-optimal masking rate by maximizing

$$P_S = (k+1)\mathbb{E}_{t \sim U[t_0, t_1]}\left[t(1-t)^k\right].$$

The maximum is reached when

$$t_0 = t_1 = \arg\max_t t(1-t)^k.$$

Set the derivative of $t(1-t)^k$ to be 0, we have

$$\frac{\mathrm{d}\left(t(1-t)^k\right)}{\mathrm{d}t} = (1-t)^k - kt(1-t)^{k-1} = 0.$$

Thus, we get $t^* = \frac{1}{k+1}$.

Notably, when $k \to \infty$, we have

$$\lim_{k \to \infty} P_S = \lim_{k \to \infty} \left(1 - \frac{1}{k+1}\right)^k = \frac{1}{e}.$$

$\square$

## F. Experimental Details

### F.1. Learning Dynamics with Uniform Attention

The model is trained using a masking range of $t \in [0, 0.2]$ with loss in Equation (1). We employ full-batch training with a batch size of 5,000, a learning rate of $1 \times 10^{-3}$, and a hidden dimension of $D = 2560$ (embedding dimension $d = 63$). As shown in Figure 5, the model transitions smoothly from random guessing (0.5 accuracy) to near-perfect generalization without the prolonged "plateau" phase typically associated with grokking in standard architectures. This suggests that the MD objective inherently stabilizes the learning of sparse functional dependencies, even when the attention structural is removed.

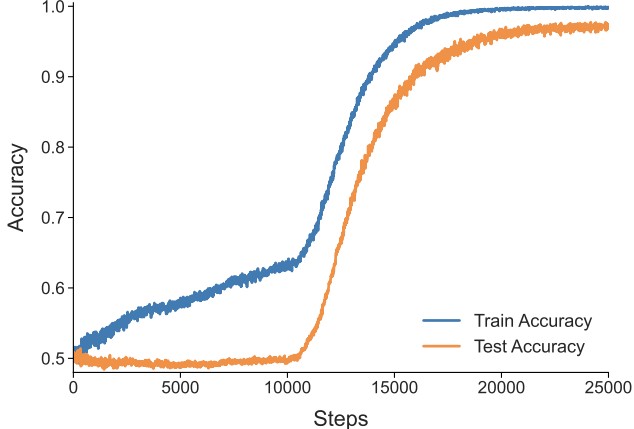

*Figure 5.* **Learning Dynamics with Uniform Attention.** Training and test accuracy on the $(20, 6)$ parity task using an MLP model where attention is fixed to a uniform distribution. The model demonstrates efficient generalization to the test set shortly after the training loss begins to converge, showing no characteristic "grokking" delay despite the absence of attention mechanisms.

### F.2. Learning parity by output supervision

In Section 4.5, we compared the convergence speeds of standard training and masked diffusion on the $(n, k) = (20, 6)$ parity problem on the nanoGPT implementations where layer norm, residual connection and cross-entropy loss are included. Figure 2 (main text) shows that within the initial training window, standard training reaches perfect training accuracy but fails to generalize, staying at the $50\%$ chance level.

To confirm that this behavior constitutes grokking rather than a total failure to learn, we extended the training duration for the standard model. As illustrated in Figure 6, the validation accuracy for the standard training objective eventually converges to $100\%$, but only after a significantly longer period of plateauing compared to the Masked Diffusion.

### F.3. Full Test Loss Trajectories Across Masking Intervals

To provide a more granular view of the training dynamics, we present the test loss trajectories for all ten sub-interval models and the full-range baseline ($\mathcal{U}[0, 1]$) in Figure 7. While the final performance in Figure 3 summarizes the outcome at 6,000

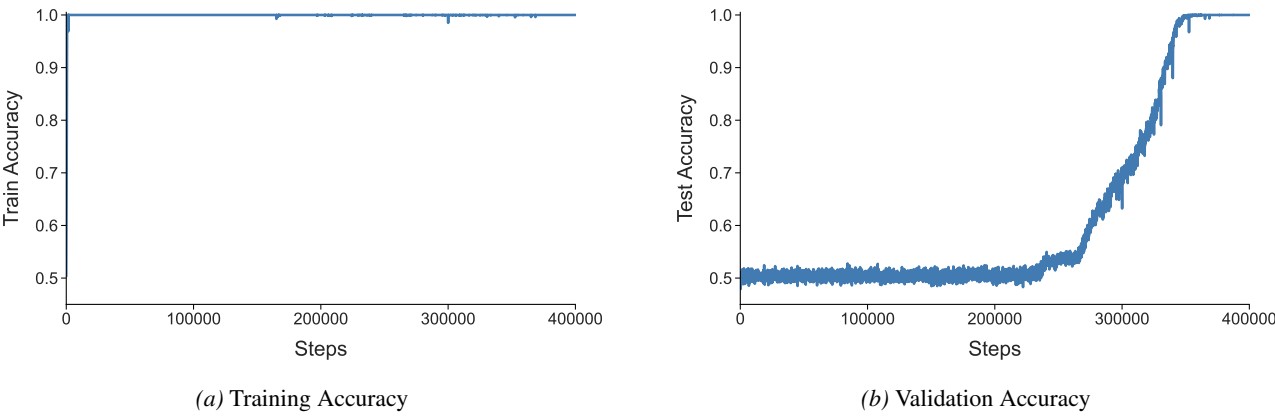

*(a)* Training Accuracy
*(b)* Validation Accuracy

*Figure 6.* **Learning parity by output supervision.** (a) Training accuracy and (b) Validation accuracy. Output supervision finally converges to $100\%$ test accuracy after a long training.

steps, the temporal data reveals that models trained in the "Signal-Rich" window ($t \in [0.4, 0.6]$) exhibit not only lower final loss but also a consistently steeper rate of convergence from the earliest stages of training.

Notably, the extreme intervals ($t \in [0, 0.1]$ and $t \in [0.9, 1.0]$) quickly plateau at a significantly higher loss level, confirming that these regions provide insufficient or overly noisy gradients for effective representation learning. The $\mathcal{U}[0, 1]$ baseline (black line) remains competitive but is consistently surpassed by the mid-range intervals, justifying our decision to concentrate the sampling budget on the most informative noise levels.

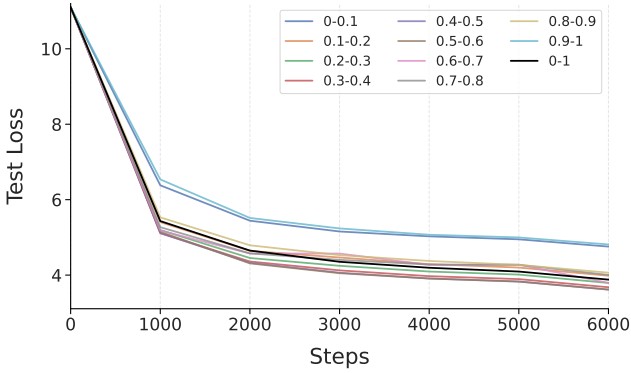

*Figure 7.* **Full Test Loss Trajectories Across Masking Intervals.** Comparative evaluation of test loss over 6,000 training steps for 50M-parameter models. Each colored line represents a model trained exclusively on a restricted masking interval of width 0.1. The black line represents the standard $\mathcal{U}[0, 1]$ baseline. Intervals centered around $t = 0.5$ show the most rapid and sustained decrease in loss, while extreme masking ratios (top lines) fail to converge to competitive minima. Zoom in for more details.

### F.4. Why Does the Optimum Fall Near $t \approx 0.5$?

The U-shaped curve in Figure 3 is not merely empirical. We now show that the location of the signal-rich window can be predicted from corpus statistics, by modeling natural language as a long-tail mixture of $n$-gram dependencies and asking which masking ratio maximizes the expected learning signal.

**Mixture of dependencies.** Natural language is well approximated by a hierarchy of $k$-gram statistics (Shannon, 1951; Jurafsky & Martin, 2026). We treat the corpus as a mixture in which a fraction $w_i$ of the predictive structure is carried by order-$i$ dependencies. Mastering the dependency of a full $i$-gram requires isolating the *synergistic* information absent from every lower-order sub-context; under Partial Information Decomposition, this pure synergy is canonically represented by the parity (XOR) interaction (Rosas et al., 2019; Williams & Beer, 2010). This grounds our use of parity as a proxy for high-order language modeling, rather than a heuristic.

**Aggregate signal and the predicted optimum.** Summing over the mixture gives the expected total learning signal

$$\text{Signal}(t) \; \propto \; \sum_i w_i \, t \, (1-t)^{\, i-1}.$$

To calculate the optimal $t$, we first estimate $w_i$ from the actual $n$-gram ($i \in [2,6]$) frequency distribution of `WikiText`, discarding $n$-grams occurring fewer than five times (Mikolov et al., 2013). The resulting long-tail weights are $w_2 = 0.7427$, $w_3 = 0.2063$, $w_4 = 0.0401$, $w_5 = 0.0084$, and $w_6 = 0.0024$. Substituting these into $\text{Signal}(t)$ and maximizing yields an optimal masking ratio of $t^\star \approx 0.46$, in close agreement with the empirically optimal window $t \in [0.4, 0.6]$ identified above. The theory therefore pinpoints the signal-rich window directly from corpus statistics, explaining *why* our chosen range $t \in [0.45, 0.55]$ is near-optimal.

We note that this estimate is only an approximation: it models the learning signal as *linear* in $P_S$. If we assume the signal scale quadratically with $P_S$ according to Theorem 4.5, $t^\star$ would shift further toward $0.5$ (to $t^\star \approx 0.48$ in the quadratic case).

### F.5. Details on dataset `tulu-3-sft-personas-math-filtered`

The `tulu-3-sft-personas-math-filtered` dataset is a refined subset of the `tulu-3-sft-personas-math` collection within the `tulu-3-sft-mixture`. It comprises 80.5k synthetically generated examples specifically designed to bolster the model's proficiency in resolving sophisticated and high-difficulty mathematical word problems.

### F.6. Details on Pretraining and Fine-tuning

We utilize the **AdamW** optimizer for both the pretraining and fine-tuning stages. To ensure the reliability of our empirical results, we conducted extensive preliminary evaluations across various scales—from toy settings to 50M parameter models—and observed a consistent performance pattern that persists at the 8B scale.

Experiments was conducted on **8×NVIDIA H100 GPUs**. Pretraining on `LLaDA-8B Base` for 15,000 steps requires approximately 20 hours per run, while fine-tuning for 1,200 steps takes roughly 6 hours. Regarding evaluation, we strictly follow the protocols established by Nie et al. (2025b), with the exception of conditional generation benchmarks, where we fix both the generation length and the number of sampling steps to 512. The specific hyperparameter configurations are summarized in Table 4.

*Table 4.* Hyperparameters for Pretraining and Fine-tuning on `LLaDA-8B Base`.

| Hyperparameter | Pretraining | Fine-tuning |
|---|---|---|
| Optimizer | AdamW | AdamW |
| Max Learning Rate | $1 \times 10^{-4}$ | $2 \times 10^{-5}$ |
| Weight Decay | 0.01 | 0 |
| Warm-up Steps | 2,000 | 160 |

