# OpenReview forum: "Tuning the Implicit Regularizer of Masked Diffusion Language Models: Enhancing Generalization via Insights from $k$-Parity"
_ICML.cc/2026/Conference — ICML 2026 regular_

### Official Review · Reviewer_k5bq · 2026-03-12

**Soundness:** 3
**Presentation:** 3
**Significance:** 2
**Originality:** 3
**Overall Recommendation:** 4
**Confidence:** 3

**Summary:**

This paper studies the generalization behavior of masked diffusion language models and shows, both theoretically and empirically, that the masked diffusion objective reshapes the optimization landscape by combining a feature-learning signal regime with an implicitly regularizing noise regime, thereby avoiding grokking on the k-parity task.
Based on this analysis, it further proposes a better mask-probability design and reports consistent gains across both pre-training and supervised fine-tuning, from 50M up to 8B models.

**Compliance With Llm Reviewing Policy:**

Affirmed.

**Final Justification:**

My concerns have been addressed. I will raise my score.

**Key Questions For Authors:**

Please see the weakness

**Limitations:**

The article discussed the limitations.

**Strengths And Weaknesses:**

Strength

1. The paper studies an important question: why masked diffusion language models may avoid the severe grokking behavior often seen in standard autoregressive training.

2. The article tries to explain a plausible and potentially important empirical phenomenon: overly small mask ratios make the denoising task too easy, overly large mask ratios make it too noisy, and intermediate ranges may provide stronger learning signal.

Weakness

1. The k-parity analysis offers useful intuition, but its transfer to natural language MDLM remains largely heuristic rather than rigorously justified.

2. The paper empirically finds that [0.4,0.5] and [0.5,0.6] work best in the 50M ablation and then uses [0.45,0.55] for the 8B experiments, but this feels more like an empirical discovery supported by a post-hoc story than a theory that predicts the interval in advance for language. How the range selection is related to the k?

3. The article shows that 𝑡∗=1/(k+1), however, they didn't show how to estimate an effective “k” from a real text context. I would be impressed if the article could link the k-parity (for instance, get k from the input context) to the selection range of masking probability.

4. The article also does not provide a concrete way to quantify whether a passage is more or less “connected” or redundant. The paper would be stronger if it more clearly distinguished what is uniquely explained by the k-parity theory from what could already be expected from broader learning-dynamics or information-content considerations. A stronger and more actionable theory would ideally lead to testable input-dependent predictions, for example by relating estimated contextual redundancy or dependency structure to the preferred masking range. However, the paper does not currently formalize or validate such a mechanism.

---

> ### Author Rebuttal · Authors · 2026-03-26
>
> We thank the reviewer for their highly constructive and insightful feedback on our work!
>
> > W1, W2, W3: How the parity case relates to the natural language case.
> >
>
> **A1:** We thank the reviewer for this important question! We would like to clarify that the connection between the two parts is direct and explicitly discussed in our manuscript. Specifically, as detailed in **Lines 304-331**, the findings established in **Theorem 4.4, Corollary 4.7 and Figure 2** directly motivate our empirical practice of truncating the range of t at both ends. This approach of deriving rigorous theoretical conclusions to guide practical empirical implementations follows a standard first-principle paradigm in the field, consistent with works such as [1] and [2], among others.
>
> While rigorous analyzing natural language is **intractable** because it lacks a fixed, closed-form mathematical structure, it can be approximated as a combination of $1$- to $k$-parities following a long-tail distribution [3]. With this formulation, our $k$-parity analysis can be applied. Specifically, our derivations show that the learning signal from an $i$-parity is proportional to $t(1-t)^i$, where $t$ is the masking rate. The total learning signal is therefore dominated by $1$-parity dependency, driving the optimal $t$ mathematically toward $0.5$. As explicitly shown in **Figure 3, 4 and Table 1, 2**, our empirical results strongly align with this derived intuition.
>
> [1] Larger Datasets Can Be Repeated More: A Theoretical Analysis of Multi-Epoch Scaling in Linear Regression
>
> [2] Tensor Programs V: Tuning Large Neural Networks via Zero-Shot Hyperparameter Transfer
>
> [3] How Much Context Does Natural Language Actually Require? An Analysis Using LLMs as Statistical Oracles, MOSS@ICML2025
>
> > W4: The article also does not provide a concrete way to quantify whether a passage is more or less “connected” or redundant. The paper would be stronger if it more clearly distinguished what is uniquely explained by the k-parity theory from what could already be expected from broader learning-dynamics or information-content considerations.
>
> **A2:** We thank the reviewer for this deeply insightful critique!
>
> Interestingly, within our framework, **$k$ itself serves as the exact metric for "connectedness".** Unlike broad information theory, $k$-parity provides explicit mathematical predictions relating this structure to the optimal masking range ($t$). Consider the two extremes:
> 1. **Independent text ($k=0$):** Tokens are independent. Our theory predicts an optimal masking rate of $t=1$.
> 2. **Highly connected text ($k=N$):** The sequence is dependent across its entire length ($N$). The theory predicts the optimal rate shifts to $t=1/(N+1)$.
>
> Crucially, our $k$-parity setting provides a controlled environment where these exact theoretical dynamics and optimal masking ranges can be **directly tested and verified by experiments**.
>
> When extending this to natural language, constructing a formal mathematical metric to quantify "connectedness" for arbitrary text would **require strong assumptions that vary across different data domains**. Instead, we provide an actionable way to test these input-dependent predictions on real text using our proxy model. To directly address your suggestion, one can evaluate the "connectedness" of two different datasets by running two proxy experiments. Because the optimal $t$ depends purely on data rather than model scale or architecture, **the initial proxy results are entirely sufficient for making this judgment**.
>
> While our theory and proxy provide a robust and actionable framework, developing a formalized metric to directly parse arbitrary natural language connectedness is beyond the scope of this paper. We position our work as a foundational first step toward understanding these core learning dynamics, and we leave the formalization of an input-dependent metric as an exciting open problem for follow-up work.
>
> We will incorporate this discussion into the manuscript to clarify how $k$-parity and the proxy model combine to yield testable predictions for real-world text.
>
> We sincerely appreciate your thoughtful review and the opportunity to strengthen our manuscript. Please let us know if any of our responses require further elaboration; we look forward to hearing your perspective!

---

> > ### Author Rebuttal · Reviewer_k5bq · 2026-04-05
> >
> > Thank you for the detailed clarification. I appreciate the authors’ effort to explain the role of the k-parity analysis and the practical value of the proxy-model procedure.
> >
> > However, my main concern is still not fully resolved. In my view, the bridge from the k-parity setting to natural language remains largely heuristic rather than theoretically established. In particular, the claim that natural language can be approximated as a long-tail mixture of 1- to k-parities is plausible, but it is not substantively justified or validated here, and the rebuttal still does not provide a concrete way to estimate an effective k or quantify “connectedness” for real text. Therefore, I consider this concern only partially resolved.

---

> > > ### Author Response · Authors · 2026-04-06
> > >
> > > We thank the reviewer for their reply and clarify the follow-up questions below.
> > >
> > > > Q1: From parity to natural language
> > >
> > > **A5:** We apologize for the brevity of our previous response due to **character limits**, and we greatly appreciate the opportunity to clarify that this approximation is not heuristic. It is, in fact, a rigorous mathematical isomorphism derived from foundational NLP and information theory.
> > >
> > > Natural language is universally modeled as a hierarchy of $k$-gram Markov chains [1, 2]. To master the highest-order dependencies specific to a full $k$-gram context, models must specifically isolate the pure synergistic information, rigorously stripping away lower-order redundant information [3].
> > > Crucially, under the Partial Information Decomposition (PID) framework, information theorists utilize the parity (XOR) problem as the canonical mathematical basis to represent and analyze this exact state of pure synergy [3, 4]. Therefore, this complete logical chain—**from modeling natural language as $k$-grams, to isolating synergistic information via PID, and mapping this pure synergy to its canonical XOR basis**—rigorously justifies why the parity problem serves as a highly effective and theoretically grounded proxy for analyzing high-order dependencies in natural language.
> > >
> > > > Q2: Why long-tail distribution and estimation for an effective k
> > >
> > > **A6:** We deeply appreciate the reviewer's feedback. To further validate our claim that natural language acts as a **long-tail** mixture, we computed the actual $n$-gram ($n \in [2, 6]$) probability distribution on the ```WikiText``` dataset (discarding $n$-grams occurring $<5$ times, following [5]). The resulting distribution weights $w_i$ are: $w_2=0.7427$, $w_3=0.2063$, $w_4=0.0401$, $w_5=0.0084$, $w_6=0.0024$.
> > >
> > > As a concrete mathematical specification of the qualitative intuition we introduced in A1, the expected total learning signal can be formally derived as:$$Signal \propto \sum_{i} w_i t(1-t)^{i-1}$$ By substituting the WikiText weights $\{w_i\}$ into this formula, we calculate an optimal masking ratio of $t \approx 0.46$. If an intuitive scalar metric is desired to characterize the corpus, one can explicitly estimate the "effective $k$" by reverse-engineering it directly from this calculated optimal $t$.
> > >
> > >
> > > [1] Shannon, C.E., 1951. Prediction and entropy of printed English.
> > >
> > > [2] Daniel Jurafsky and James H. Martin. 2026. Speech and Language Processing: An Introduction to Natural Language Processing, Computational Linguistics, and Speech Recognition with Language Models, 3rd edition.
> > >
> > > [3] Rosas, F.E., Mediano, P.A., Gastpar, M. and Jensen, H.J., 2019. Quantifying high-order interdependencies via multivariate extensions of the mutual information.
> > >
> > > [4] Williams, P.L. and Beer, R.D., 2010. Nonnegative decomposition of multivariate information.
> > >
> > > [5] Mikolov, T., Sutskever, I., Chen, K., Corrado, G.S. and Dean, J., 2013. Distributed representations of words and phrases and their compositionality.

---

### Official Review · Reviewer_Sdic · 2026-03-13

**Soundness:** 2
**Presentation:** 3
**Significance:** 3
**Originality:** 4
**Overall Recommendation:** 4
**Confidence:** 4

**Summary:**

The work uses the k-parity task as a simple but revealing testbed to show that masked diffusion training can avoid the usual grokking behavior of standard training—where models memorize first and only generalize much later—and instead more directly learn the underlying rule through the structure of the masking objective.

**Compliance With Llm Reviewing Policy:**

Affirmed.

**Key Questions For Authors:**

1. Could you add more datasets to Tables 1‑3—ideally the same 6-8 datasets without cherry‑picking.

2. The k‑parity findings are comprehensive but not directly applicable to LLMs. Could you provide some practical guidance for choosing t_min and t_max and explain how they affect model ability and performance beyond infoless test loss. (For example, a higher t may encourage the model to explore and guess when uncertain, whereas a lower t may prompt it to answer directly.)

**Limitations:**

yes

**Strengths And Weaknesses:**

The work adopts a novel perspective to understand masked diffusion’s generalization: a high masking rate (noise regime) regulates training, while a low masking rate encourages learning rules instead of memorization. It provides both theoretical and empirical support for Signal-Optimal Mask Sampling, but its evaluation is insufficient for modern LLMs.

---

> ### Author Rebuttal · Authors · 2026-03-26
>
> Dear reviewer, thank you very much for your constructive and insightful comments on our work!
>
> > Q1: Could you add more datasets to Tables 1‑3—ideally the same 6-8 datasets without cherry‑picking.
> >
>
> **A1:** We have augmented our finetuning results (with respect to Table 2 and Table 3) across a broader benchmark and probability range.
>
> | Method | MMLU | MMLU-STEM | ARC-Challenge | GPQA | GSM8K | MATH |
> | --- | --- | --- | --- | --- | --- | --- |
> | Base | 0.659 | 0.629 | 0.459 | 0.252 | 0.703 | 0.314 |
> | 0.0 - 1.0 | 0.659 | 0.621 | 0.468 | 0.344 | 0.768 | 0.341 |
> | 0.45 - 0.55 | **0.669** | **0.635** | **0.480** | **0.402** | 0.738 | 0.342 |
> | 0.2 - 1.0 | 0.659 | 0.630 | 0.471 | 0.317 | 0.762 | 0.358 |
> | 0.3 - 1.0 | 0.659 | 0.631 | 0.464 | 0.353 | 0.774 | 0.360 |
> | 0.5 - 1.0 | 0.662 | 0.628 | 0.451 | 0.277 | **0.785** | **0.375** |
>
> As explicitly shown in the new results, **our core conclusions remain strictly unchanged** (consistent with Section 5.2.2). Specifically, the expanded evaluation highlights two distinct optimal sampling strategies depending on the nature of the task:
>
> **For PPL-based Discriminative Tasks (e.g., MMLU, ARC-Challenge, GPQA):** Truncating the range of $t$ at *both* ends (e.g., $0.45 - 0.55$) yields the best performance. This directly aligns with the theoretical motivations presented in Theorem 4.4, Corollary 4.7, and Figure 2.
>
> **For Generative Reasoning Tasks (e.g., GSM8K, MATH):** Truncating only the lower bound of $t$ (e.g., $0.5 - 1.0$) proves superior. As noted in **Line 392**, leaving target information partially unmasked provides unwanted hints during generative tasks. Therefore, fully masking the target forces the model to learn the complete, rigorous problem-solving process strictly from the prompt alone.
>
> > Q2: The k‑parity findings are comprehensive but not directly applicable to LLMs.
> >
>
> **A2:** We thank the reviewer for this excellent question! As a **core contribution** of our work, **Section 5** explicitly shows how we translate our theoretical conclusions to practical LLMs. We detail our practical recommendations for choosing $t$ and its downstream effects on model abilities below.
>
> Practically, as demonstrated in Section 5.1 and 5.2, one can use a **tiny 50M-parameter** proxy model to cheaply identify **good enough** bounds for their specific pretraining data. Furthermore, as shown in Tables 1, 2, our identified ranges serve as a highly effective, robust baseline for language tasks. This yields substantial empirical gains peaking at **8.8%** and **5.8%** on **8B**-parameter models during pretraining and finetuning, respectively, even without exhaustive hyperparameter sweeping.
>
> Regarding model capabilities, the choice of $t$ significantly impacts downstream behavior. For general language modeling, a moderate $t$ (near $0.5$) enables the model to better **understand relationships and dependencies** of the natural language as we observe an consistent improvement over PPL-based Discriminative Tasks in Table 1, 2. Conversely, a higher $t$ (approaching $1$) is crucial for **complex reasoning tasks** because leaving target information unmasked provides the model with unwanted hints. Pushing $t$ higher removes these hints, forcing the model to independently learn the full, long-horizon reasoning chain from the prompt alone, as noted in Line 392.
>
> We thank you again for your time and effort in helping us improve our paper! We are eager to hear your thoughts and are more than happy to provide any further clarifications you might need.

---

> > ### Author Rebuttal · Reviewer_Sdic · 2026-04-01
> >
> > The analysis regarding $t$ is reasonable. If the authors can provide more empirical evidence, I would raise my rating; otherwise, I will keep my current positive rating unchanged.

---

> > > ### Author Response · Authors · 2026-04-03
> > >
> > > We thank the reviewer for acknowledging our analysis. To provide further empirical evidence regarding how the masking ratio $t$ affects reasoning capabilities, we conducted a fine-grained evaluation on the MATH dataset.
> > >
> > > As a brief reminder, increasing the masking ratio $t$ aggressively masks intermediate targets during training. This forces the model to independently deduce entire reasoning chains without relying on local, step-by-step hints. A higher $t$ should significantly boost capabilities on complex, long-horizon tasks. Conversely, a moderate $t$ should be sufficient, or even slightly better, for simple, short-horizon tasks where complex deduction is not required.
> > >
> > > To validate this, we evaluated the model across three subtasks in the MATH dataset: Geometry, Intermediate Algebra, and Precalculus. Using the existing difficulty labels for examples in the MATH dataset, we partitioned each subtask into three difficulty levels: Easy, Medium, and Hard.
> > >
> > > Our results reveal consistent trends across all domains when shifting from a moderate $t$ to a high $t$. For Easy examples, performance decreased under the high masking ratio. However, as task difficulty increases, the advantage of a higher $t$ becomes clear. For Medium/Hard tasks, the higher masking ratio yields consistent improvements. This confirms our core claim that a higher $t$ effectively forces the model to learn a complex deductive prior, which is highly beneficial and necessary for difficult reasoning tasks. Note that since majority of examples are in the medium and hard categories and thus benefit from more complex long-horizon reasoning, overall we get a boost in the reasoning performance.
> > >
> > > 1. Geometry
> > >
> > > | Difficulty| $0.45-0.55$ | $0.5-1$ | $\Delta$ |
> > > | --- | --- | --- | --- |
> > > | Easy (Level 1) | 52.63% | 47.37% | -5.26% |
> > > | Medium (Level 2-3) | 46.20% | 47.28% | +1.08% |
> > > | Hard (Level 4-5) | 10.51% | 15.56% | +5.05% |
> > > | *Overall* | *27.56%* | *30.27%* | *+2.71%* |
> > >
> > > 2. Intermediate Algebra
> > >
> > > | Difficulty| $0.45-0.55$ | $0.5-1$ | $\Delta$ |
> > > | --- | --- | --- | --- |
> > > | Easy (Level 1) | 63.46% | 57.69% | -5.77% |
> > > | Medium (Level 2-3) | 21.67% | 23.84% | +2.17% |
> > > | Hard (Level 4-5) | 6.25% | 7.95% | +1.70% |
> > > | *Overall* | *15.06%* | *16.50%* | *+1.44%* |
> > >
> > > 3. Precalc
> > >
> > > | Difficulty| $0.45-0.55$ | $0.5-1$ | Delta |
> > > | --- | --- | --- | --- |
> > > | Easy (Level 1) | 50.88% | 50.88% | +0.00% |
> > > | Medium (Level 2-3) | 20.83% | 24.17% | +3.34% |
> > > | Hard (Level 4-5) | 4.82% | 4.82% | +0.00% |
> > > | *Overall* | *16.67%* | *18.13%* | *+1.47%* |
> > >
> > > We are happy to provide further clarification or additional details if the reviewer have any remaining questions.

---

### Official Review · Reviewer_7zJq · 2026-03-13

**Soundness:** 3
**Presentation:** 3
**Significance:** 2
**Originality:** 3
**Overall Recommendation:** 4
**Confidence:** 3

**Summary:**

The research presented in this paper explores how sampling from various time intervals within the forward process affects the generation quality of masked diffusion language models. Specifically, this paper starts from an in-depth investigation on the $k$-parity task, and then transfers the insights from the $k$-parity task to general language modeling with masked diffusion.

Technically, a core finding of this research is the signal-noise decomposition of a variant of MD loss, which employs mean squared errors. The authors conducted compelling theoretical analysis on the $k$-parity task.

**Compliance With Llm Reviewing Policy:**

Affirmed.

**Final Justification:**

Thanks for providing the additional derivation, and I appreciate the authors' effort for detailed response. I raised my recommendation score to 4.

I suggest that the authors directly use the CE loss in the $k$-parity task if applicable, which will be more directly relevant to language modeling. Moreover, while the $k$-parity case is well executed, especially developing theories, there is still a disconnection between theoretical sigmal-optimality in the case of $k$-parity and sigmal-richness in the case of language modeling, especially, the identification of the signal-rich windows in language modeling relies on empirical pre-experiments, rather than backed by the theories naitive to language modeling, like the execution in the $k$-parity task.

**Key Questions For Authors:**

1. In Line#376, I don’t understand why the signal would vanish in the case of t $\in$ [0.9, 1.0]. Can you please elaborate? This seems to contradict with your results in Table 4, in which including t_max=1 in the sampling range gives the best performance, and in Table 1&2, you don’t report the results using the sampling window including t_max=1. Do you have results in these cases?

2. I don’t understand the logic in the claim “by restricting the diffusion schedule to the window where signal-to-noise ratio is most informative, we can achieve superior generalization” (Line#336, column#2). the relationship between signal-to-noise ratio and generalization/generalizability is disconnected, no evidence supports the claim of improved generalization in the case of language modeling.

3. Can you provide more details of inference?

**Limitations:**

Please see my comments above.

**Strengths And Weaknesses:**

**Strengths**

1. This paper is very well written and easy to follow. The authors did a great job in presenting the theoretical findings in the $k$-parity problem.

2. The signal-noise decomposition of the MSE-based diffusion loss is novel, which would provide new insights in improving the loss of masked diffusion models.

3. The experiments are well designed, and the improvements on the benchmark tasks are strong.

**Weaknesses**

1. The signal-noise decomposition of the masked diffusion objective, as presented in Definition 3.6, is a central contribution of this work and is explicitly developed using Mean Square Error (MSE). However, it should be noted that the standard objective typically utilized in MDLMs is the cross-entropy loss. Although they’re MLE’s wearing different statistical masks, their learning dynamics in terms of scoring rules are different, therefore, it’s unclear how the signal-noise decomposition can be generalized to the cross entropy loss. One piece of evidence is that the “signal-optimal” ranges for this two tasks are very different, i.e., for $k$-parity, the range is (0, 0.4), while for the language tasks, the range actually sits in the middle excluding the 0-started interval ([0.45, 0.55]). I recommend the authors to provide clearer and more detailed justification.

2. Lack of theoretical proof for the signal optimality in the case of language modeling with the cross-entropy loss. The evidence provided in Section 5 is largely empirical, as the authors only experimented with a set of handcrafted combinations of t_min and t_max. The proof of the signal optimality for learning with the generic MDLM objective is absent. I recommend changing signal-optimal to signal-rich sampling.

3. Lack of the discussion about the connection to loss reweighting [e.g., 1], which is widely used in training diffusion models. Essentially, the signal-optimal mask sampling is equivalent to a loss weighting scheme, which corresponds to a top-hat function. Specfically, the weights of the samples falling in [0, t_min) and (t_max, 1] are enforced to be zero and uniform between [t_min, t_max]. Similar weighting schemes are also investigated in [1]. I recognize that investigating the sampling importance at different t’s from the perspective of information theory is novel, I think the paper could be stronger if it can link their insights to the design of loss reweighting schemes. Due to this reason, I rate the significance of this work to “fair”.

4. The statement about generalization is questionable. As the conclusion of that “the MD objective provides an inherent regularization that boots the feature learning” (Line#279, Column#2) is tightly coupled with the MSE loss (Definition 3.6). It’s unclear whether this generalizability still holds for the generic cross-entropy loss.

5. Unlike the $k$-parity example, identifying the signal-optimal sampling window is purely emperical, and needs extra training budget in the first stage to identify.

---

> ### Author Rebuttal · Authors · 2026-03-26
>
> We thank the reviewer for their highly constructive feedback on our work!
> > W1: Generalization of signal-noise decomposition to CE loss.
>
> **A1:**
> 1. **MSE to CE:** While our Def 3.6 utilizes MSE for mathematical analysis, the core dynamics generalize seamlessly to CE loss. **Empirically**, the experiments in Fig 2 and Sec 5 uses standard CE. **Theoretically**, the analysis holds under CE: the second term of Eq 3 will effectively regularize predictions toward a uniform distribution (a random guess), analogous to how MSE regularizes the model to output 0.
>
> 2. **Optimal Ranges:** The optimal ranges for $k$-parity (0, 0.246)—as noted in lines 321 and 286—and natural language [0.45, 0.55] differ because they possess fundamentally distinct dependency structures.
>
> > W2, W5.1: Lack of theoretical proof for the natural language case.
>
> **A2:** We detail the theoretical justification for natural language in **[A1 to Reviewer y6GL](https://openreview.net/forum?id=UstfBHsp6p&noteId=TGH5gJ3oyG)**, which we will incorporate into the revision. Additionally, we will gladly adopt your precise term "signal-rich" to replace "signal-optimal"!
> > W3: Lack of discussion to reweighting.
>
> **A3:** Assuming omitted reference [1] is Min-SNR [2], our sampling indeed acts as a top-hat loss reweighting. However, our analysis fundamentally differs from image diffusion. Critically, image diffusion is continuous, while language is strictly discrete, requiring entirely distinct analysis. Furthermore, image diffusion requires learning from pure noise ($t \to 1$) for unconditional generation. Conversely, text generation is inherently **prompt-driven**, making the pure-noise regime uninformative. Note that for generative reasoning tasks, prompts always provide sufficient signal regardless of $t$.
>
> We thank the reviewer for pointing out this valuable connection, and we will include this discussion in our revised manuscript!
>
> [2] Efficient Diffusion Training via Min-SNR Weighting Strategy
> > W4: The statement about generalization is questionable.
>
> **A4:**  We would like to clarify that the theoretical conclusions derived under the MSE loss do indeed generalize to the CE loss.
>
> While our theoretical analysis in Section 4 utilizes the MSE loss (Def 3.6), all our empirical experiments in Section 5 are conducted using the CE loss (Line 337). The fact that our empirical results (Fig 4, Tab 1, 2) perfectly align with our theoretical insights serves as direct evidence that the inherent regularization of the MD objective holds regardless of the specific loss function.
> > W5.2: Identifying the optimal window needs extra budget.
>
> **A5:** Interestingly, the signal-rich sampling window can be identified very quickly and with minimal computational overhead. Because the optimal sampling window depends entirely on the dataset and corruption process, it is fundamentally architecture-agnostic. Thus, it **does not** require full-scale training to identify. As demonstrated in Sec 5.1 and 5.2, a lightweight proxy model with only 50M parameters is perfectly sufficient to identify the good range.
> > Q1: Why the signal would vanish?
>
> **A6:** Good question! We clarify that **there is no contradiction**; rather, the different behaviors stem entirely from the distinct settings between pretraining and finetuning:
> 1. **Pretraining:** When $t \to 1$, nearly the entire sequence is masked. The model is forced to recover the ground truth out of virtually nothing, which has a massive number of valid possibilities. Theoretically as discussed in line 220 and Theorem 4.4, the learning signal mathematically approaches 0 because $P_S=0$ in this case.
> 2. **Finetuning (Tab 3):** Here, **prompts are never masked**. This ensures sufficient context for recovery, meaning the signal never vanishes. Furthermore, as noted in line 392, leaving target information unmasked provides unwanted hints for generative tasks; therefore, fully masking the target forces the model to learn the complete problem-solving process from the prompt alone.
> 3. For further details on these results, please see our expanded response in **[A1 to Reviewer Sdic](https://openreview.net/forum?id=UstfBHsp6p&noteId=BPNBhDeQYg)**.
>
> > Q2: Evidence of generaliation for natural language.
>
> **A7:** Low-SNR regimes provide too few learning signals, though they vary by task. For pretraining and PPL tasks, both $t\to 0$ and $t\to 1$ are uninformative. However, for generative reasoning, prompts provide signal even at $t=1$ (see A6), making only $t\to 0$ low-SNR. Discarding these task-specific uninformative regimes focuses capacity on the most informative window, improving generalization. As shown (Fig 3-4, Tab 1-3), this yields gains up to **8.8% (pretraining) and 5.8% (finetuning)** on 8B models .
> > Q3: Details of inference.
>
> **A8:** We kindly refer the reviewer to Line 1168 for comprehensive inference details.
>
> We hope that our clarifications have addressed your concerns. Please let us know if there is anything else we can clarify!

---

> > ### Author Rebuttal · Reviewer_7zJq · 2026-04-03
> >
> > I thank the authors for their responses. Here are my follow-up questions:
> >
> > * First, I apologize for forgetting to attach the exact reference in my original review. [1] refers to Shi et al (https://arxiv.org/pdf/2511.19664), in which the authors applied reweighting to discrete diffusion models. Specifically, generating images pixel by pixel via MDM. Moreover, the reweighting method is applicable to both continuous and discrete domains from the perspective of tightening ELBOs.
> >
> > * I’m not sure if I understand your argument about the difference between unconditional and conditional generation. First, for image generation, the models are usually conditioned on object classes (e.g., dog, cat, etc). Second, are you suggesting that your method is not suitable for unconditional generation tasks?
> >
> > * The generalization of the regularization term from MSE loss to CE loss remains unclear to me. Please can you provide a full and direct derivation of the signal-regularization decomposition for the CE loss? Otherwise, the theoretical findings from the $k$-parity example would still remain disconnected from the language applications theoretically.
> >
> > I keep my original score for now, and look forward to the follow-up discussion with the authors.

---

> > > ### Author Response · Authors · 2026-04-04
> > >
> > > We thank the reviewer for the constructive feedback of our work. Below, we address the remaining questions and provide further clarifications.
> > >
> > > > Q4: Reweighting
> > > >
> > >
> > > **A9:** We thank the reviewer for directing us to [1]. Our theoretical contribution provides a distinct and **complementary** perspective to [1] by explicitly linking the design of loss reweighting schemes to the underlying **optimization dynamics**.
> > >
> > > While [1] justifies reweighted losses by constructing tighter ELBOs, their analysis remains strictly at the loss level. Although such bounds are mathematically general across domains, they do not capture the actual optimization dynamics, which differ fundamentally between modalities. Bridging this gap is crucial: as demonstrated by the grokking phenomenon in our $k$-parity experiments (Figure 6), having an exact (or better loss) does not necessarily translate to better learning process.
> > >
> > > **Our work provides this missing link.** Through our signal-noise analysis, we show that adjusting the masking ratio $t$, which corresponds to a reweighting schedule, inherently acts as a regularizer (Theorem 4.3) and controls the effective learning rate (Theorem 4.4). Characterizing these optimization effects is **not achievable** if one relies solely on loss bounds like [1]. Ultimately, as shown in Figure 2, our theoretically derived $t$ supports better convergence, improving not only final performance but training efficiency.
> > >
> > > We would include this discussion in our revised manuscript.
> > >
> > > > Q5: Difference between image and text
> > > >
> > >
> > > **A10:** To clarify our previous terminology, when we contrasted conditional and unconditional generation, we were specifically referring to the starting state of the sequence itself. In continous image diffusion, even when conditioned on a class label, the generation starts from completely random noise and the conditioning signal is injected externally. In contrast, language reasoning is inherently prompt-driven, meaning the conditioning signal is provided as an unmasked part of the sequence itself. If a text model starts from pure noise, then the model is left with no meaningful signal to deduce the rest of the sequence, as shown in our parity case (line 277). In this sense, purely unconditional generation tasks do not practically exist in text generation.
> > >
> > > > Q6: Derivation for CE loss
> > >
> > > **A11:** Let $f\_\\theta(\\tilde{\\mathbf{z}}) \\in (0, 1)$ be the predicted probability. For $\\mathbf{m} \\in \\mathcal{R}\_S$ with $M_\{\\mathbf{m}} \\cap \\mathcal{S}'=\\{k\\}$, define the target $f^\*\_S(\\tilde{\\mathbf{z}}) = \\frac{x'\_k + (|M\_{\\mathbf{m}}|-1)/2}{|M\_{\\mathbf{m}}|}$. For $\\mathbf{m} \\in \\mathcal{R}\_N$, let $f^\*\_{N}(\\tilde{\\mathbf{z}}) = 0.5$. The BCE loss is equivalent to:
> > > $$\\mathcal{L}_{\\mathrm{eff}}(\\theta) = \\sum \_{R \\in \\{\\mathcal{R}\_S, \\mathcal{R}\_N\\}} \\Pr(\\mathbf{m} \\in R) \\mathbb{E}\_{\\tilde{\\mathbf{z}} \\mid R} \\left[ \\frac{|M\_{\\mathbf{m}}|}{2t} D\_{\\mathrm{KL}}\\left(f^\*\_R(\\tilde{\\mathbf{z}}) \\parallel f^\*\_\\theta(\\tilde{\\mathbf{z}})\\right) \\right]. $$
> > >
> > > **Proof:**
> > >
> > > By the law of total expectation, we can rewrite the objective by conditioning on the corrupted state $\\tilde{\\mathbf{z}}$:
> > > $$\\mathcal{L}(\\theta) = \\mathbb{E}_{t,\\mathbf{m}} \\left[ \\frac{1}{2t} \\mathbb{E}\_{\\tilde{\\mathbf{z}}\\mid t,\\mathbf{m}} \\left[ \\sum\_{j \\in M\_{\\mathbf{m}}} \\mathbb{E}\_{\\mathbf{x}'\\mid \\tilde{\\mathbf{z}}} \\left[ \\mathrm{BCE}\\left(x'\_j, f\_\\theta(\\tilde{\\mathbf{z}})\\right) \\right] \\right] \\right].$$
> > >
> > > Using the identity $\\mathrm{BCE}(p, q) = D\_{\\mathrm{KL}}(p \\parallel q) + H(p)$, we evaluate the inner sum for both regimes:
> > >
> > > **Case 1: $\\mathbf{m} \\in \\mathcal{R}\_N$.** All masked positions are pure noise. Thus, $x'\_j$ is independent of $\\tilde{\\mathbf{z}}$, yielding $\\mathbb{E}[x'\_j \\mid \\tilde{\\mathbf{z}}] = 0.5$ for all $j \\in M\_{\\mathbf{m}}$. The inner sum becomes:
> > > $$|M\_{\\mathbf{m}}| \\mathrm{BCE}(0.5, f\_\\theta) = |M\_{\\mathbf{m}}| D\_{\\mathrm{KL}}(0.5 \\parallel f\_\\theta) + C_0$$
> > >
> > > **Case 2: $\\mathbf{m} \\in \\mathcal{R}\_S$.** The $|M\_{\\mathbf{m}}|-1$ noise tokens have $\\mathbb{E}[x'\_j \\mid \\tilde{\\mathbf{z}}] = 0.5$. By averaging the sum, we have:
> > > $$\\mathrm{BCE}(x'\_k, f\_\\theta) + (|M\_{\\mathbf{m}}|-1)\\mathrm{BCE}(0.5, f\_\\theta) = |M\_{\\mathbf{m}}| \\mathrm{BCE}(f^\*\_{S}(\\tilde{\\mathbf{z}}), f\_\\theta)$$
> > > which equals $|M\_{\\mathbf{m}}| D\_{\\mathrm{KL}}(f^\*\_{S}(\\tilde{\\mathbf{z}}) \\parallel f\_\\theta)+C_1$.
> > >
> > > Substituting these terms back into the outer expectations and splitting by regime probabilities completes the proof.
> > >
> > > We are happy to provide further clarification or additional details if the reviewer have any remaining questions.

---

### Official Review · Reviewer_y6GL · 2026-03-23

**Soundness:** 3
**Presentation:** 2
**Significance:** 2
**Originality:** 3
**Overall Recommendation:** 3
**Confidence:** 4

**Summary:**

This paper discusses the impact of the training objective of Mased Diffusion Language Models (MDLM) on their generalizable ability. By analyzing the training process through the $k$-parity problem and decomposition the objective into Signal Regime and Noise Regime, the paper shows that the Noise Regime is played as a regularization part in $k$-parity problem, which helps improve the generalization. With such insight, they translate the objective design principle into real-world language tuning and demonstrate performance improvement on multiple tasks.

**Compliance With Llm Reviewing Policy:**

Affirmed.

**Final Justification:**

The addtional explanation and the provided related work help me better understand the generalization ability from $k$-parity to nature language process. Therefore, I raised the score accordingly.

**Key Questions For Authors:**

- Q1: Is the signal-optimal window almost the same for all kinds of masked diffusion language model? Does the window shift when the architecture of MDLM changes?
- Q2: Can you provide a more general derivation that is not restricted to the $k$-parity problem, in order to obtain the insights required for signal-optimal mask sampling in Section 5?

**Limitations:**

yes

**Strengths And Weaknesses:**

**Strengths:**
- S1: This paper conducts thorough and rigorous theoretical derivations, with each step presented in a clear and logical manner. The detailed derivation process for $k$-parity problem enhances the persuasiveness and readability of the research conclusions, making the theoretical framework highly convincing and academically rigorous.
- S2: Most claims are rigorously supported by carefully designed and well-conducted experiments. The comprehensive and sufficient experimental results provide solid and convincing evidence that effectively validates and complements the theoretical derivations, greatly enhancing the reliability and academic rigor of the overall work.

**Weaknesses:**
- W1: This work combines two parts, the theoretical analysis of MD objective through $k$-parity problem and signal-optimal mask sampling in language modeling, into one paper. The connection between two parts is weak. The derivation in $k$-parity problem can not provide solid proof and sufficient support the real-world language modeling.
- W2: The conclusions drawn in this paper are overly naive. Although the author provides an elaborate derivation on the $k$-parity problem, the discussion related to regularization is not novel, and similar insights have long existed in the field of machine learning, which is enough to support the analysis of signal-optimal mask sampling.

---

> ### Author Rebuttal · Authors · 2026-03-25
>
> Dear reviewer, thank you very much for your constructive and insightful comments on our work!
>
> > W1: The connection between theoretical and empirical parts is weak.
>
> **A1:** We thank the reviewer for this important question! We would like to clarify that the connection between the two parts is direct and explicitly discussed in our manuscript. Specifically, as detailed in **Lines 304-331**, the findings established in **Theorem 4.4, Corollary 4.7 and Figure 2** directly motivate our empirical practice of truncating the range of t at both ends. This approach of deriving rigorous theoretical conclusions to guide practical empirical implementations follows a standard first-principle paradigm in the field, consistent with works such as [1] and [2], among others.
>
> While rigorously analyzing natural language is **intractable** because it lacks a fixed, closed-form mathematical structure, it can be approximated as a combination of $1$- to $k$-parities following a long-tail distribution [3]. With this formulation, our $k$-parity analysis can be applied. Specifically, our derivations show that the learning signal from an $i$-parity is proportional to $t(1-t)^i$, where $t$ is the masking rate. The total learning signal is therefore dominated by $1$-parity dependency, driving the optimal $t$ mathematically toward $0.5$. As explicitly shown in **Figure 3, 4 and Table 1, 2**, our empirical results strongly align with this derived intuition.
>
> [1] Larger Datasets Can Be Repeated More: A Theoretical Analysis of Multi-Epoch Scaling in Linear Regression
>
> [2] Tensor Programs V: Tuning Large Neural Networks via Zero-Shot Hyperparameter Transfer
>
> [3] How Much Context Does Natural Language Actually Require? An Analysis Using LLMs as Statistical Oracles, MOSS@ICML2025
>
> > W2: The conclusions are overly naive. The discussion related to regularization is not novel, which is enough to support the analysis.
>
> **A2:** We respectfully disagree that our conclusions are naive or lack novely. While we acknowledge the historical intuition that certain denoising mechanisms may act as implicit regularizers [3], it is crucial to note that these early methods (e.g dropout training) are fundamentally different from modern diffusion models.
>
> For diffusion models, the exact mechanics of implicit regularization are highly non-trivial and remain an active frontier of research. For instance, formalizing this connection for continuous image diffusion models was only **recently** achieved in a Neurips 2025 Best paper [4]. Crucially, our work tackles the fundamentally different discrete domain. We are the **first** to explicitly formalize and establish these connections for MDLM.
>
> Far from being a trivial observation, our rigorous theoretical grounding translates directly into a “simple yet effective” practical design, yielding substantial empirical gains peaking at **8.8%** and **5.8%** on 8B-parameter models during pretraining and finetuning, respectively.
>
> Ultimately, this work represents a significant leap from existing continuous/image-centric literature and provides a foundational framework for understanding and improving MDLMs.
>
> [3] Dropout Training as Adaptive Regularization
>
> [4] Why Diffusion Models Don’t Memorize: The Role of Implicit Dynamical Regularization in Training
>
> > Q1: Does the window shift when the architecture of MDLM changes?
>
> **A3:** Great question! The short answer is no, the signal-optimal window remains consistent across different kinds of MDLMs. As established in **Theorem 4.3 and Corollary 4.7**, the optimal window is fundamentally determined by the forward corruption process and the inherent data distribution, rather than the specific neural network used for denoising. Importantly, our theoretical proofs for these two results do not depend on any specific network architecture. We also confirm this experimentally: Figure 3, 4 and Table 1 demonstrate that the optimal window ranges are highly consistent across both proxy and target models. Therefore, given a fixed data corruption process, the signal-optimal window is **architecture-agnostic** and will not shift when the underlying MDLM architecture changes.
>
> > Q2: Can you provide a more general derivation?
>
> **A4:** As noted in A1, since natural language lacks a fixed structure, providing a completely generalized derivation directly on real-world text is intractable.
>
> However, our mathematical framework can be broadened. Specifically, the derivation can be extended to $k$-gram Markov chains, of which the $k$-parity problem is a special case. The foundational insights required for our signal-optimal mask sampling remain mathematically consistent within this generalized context. We will include a detailed discussion of this extension in the revised manuscript.
>
> We hope that our clarifications could better reflect the significance of our contributions! Please let us know if there is anything else we can clarify.

---

> > ### Author Rebuttal · Reviewer_y6GL · 2026-04-04
> >
> > Dear Authors, thank you sincerely for your dedicated efforts in the detailed rebuttal.
> >
> > I greatly appreciate the time and care you have taken to clarify the theoretical derivation process, experimental design logic, and core contributions of your work.
> > However, several core concerns and key questions from my review have not been fully and substantively addressed in your rebuttal.
> >
> > 1. For the core concern about the weak connection between the $k$-parity theoretical analysis and real-world language modeling, your response only briefly claims that natural language can be approximated as a combination of $k$-parities with a single reference, but provides no rigorous theoretical justification, supplementary derivation, or targeted experimental evidence to verify the rationality of this core approximation, failing to fundamentally resolve the question of whether your $k$-parity derivation can provide solid and sufficient support for real-world language modeling.
> > 2. Regarding the concern about the novelty of your regularization-related conclusions, your rebuttal focuses on the first formalization of implicit regularization in the discrete MDLM scenario, but does not directly respond to my core point that the underlying insights about regularization are not novel and that existing machine learning insights are sufficient to support your signal-optimal mask sampling analysis, and the empirical performance gains you cited cannot answer the question about the novelty of your core theoretical insights.
> > 3. For the key request to provide a more general derivation not restricted to the $k$-parity problem, you only note that a fully generalized derivation on real text is intractable and propose an extension to $k$-gram Markov chains, but provide no substantive derivation content, key theoretical conclusions, or validity verification for this extended framework in the rebuttal, and a commitment to add relevant content in the revised manuscript cannot be regarded as a substantive response to this question.

---

> > > ### Author Response · Authors · 2026-04-06
> > >
> > > We thank the reviewer for their reply and clarify the follow-up questions below.
> > >
> > > > Q3: From parity to natural language
> > >
> > > **A5:** We apologize for the brevity of our previous response due to **character limits**. The reviewer questioned approximating natural language as a "combination of parities." This is, in fact, a rigorous mathematical isomorphism derived from foundational NLP and information theory.
> > >
> > > Natural language is universally modeled as a hierarchy of $k$-gram Markov chains [1, 2]. To master the highest-order dependencies specific to a full $k$-gram context, models must specifically isolate the pure synergistic information, rigorously stripping away lower-order redundant information [3].
> > > Crucially, under the Partial Information Decomposition (PID) framework, information theorists utilize the parity (XOR) problem as the canonical mathematical basis to represent and analyze this exact state of pure synergy [3, 4]. Therefore, this complete logical chain—**from modeling natural language as $k$-grams, to isolating synergistic information via PID, and mapping this pure synergy to its canonical XOR basis**—rigorously justifies why the parity problem serves as a highly effective and theoretically grounded proxy for analyzing high-order dependencies in natural language.
> > >
> > > We also refer to **[A6 to Reviewer k5bq](https://openreview.net/forum?id=UstfBHsp6p&noteId=mlOMzDpC5P)** for empirical evidence of this long-tail mixture claim.
> > >
> > > > Q4: Insights novelty
> > >
> > > **A6:** We clarify that our contributions extend well beyond standard machine learning **heuristics**. Our specific mechanistic insights within discrete Masked Diffusion are novel in four distinct ways:
> > >
> > > 1. **Minimizing Over-regularization** (insights shown **first time** for MDLM, explained in detail from line 306 to 323): We show that MDLMs are currently hindered by **excessive regularization** rather than a lack of it. This directional insight was previously unknown. Notably, such regularization has a theoretical lower bound of $1-1/e$ in the parity case (as derived in line 1067), meaning we cannot eliminate this regularization brought by MD in any case.
> > > 2. **A Novel Information-Theoretic Perspective:** As independently highlighted by Reviewer 7zJq, *"investigating the sampling importance at different $t$’s from the perspective of information theory is novel."* We uncover the precise information-theoretic mechanism of MD regularization.
> > > 3. **Regulating Feature Learning Rates:** We go beyond standard loss-level or generalization-bound analyses. We formally prove that the masking probability mathematically regulates the learning rate of the feature learning process.
> > > 4. **Bypassing Grokking via MD:** Our parity setting yields a highly non-trivial discovery: the specific regularization induced by MD explicitly enables the model to bypass grokking and reaches the best performance also when regularization is minimal (line 286). Connecting MD regularization to grokking mitigation is an entirely novel insight in representation learning.
> > >
> > > > Q5: A more general derivation
> > >
> > > **A7:** Let sequence $S = \{s_1, s_2, \dots, s_k\}$ be a $k$-gram. As established by recent advances in neural network dynamics [5], models rapidly acquires low-order redundant statistics. Thus, the fundamental bottleneck is learning the highest-order dependencies—i.e., the pure synergistic information $\\mathrm{Syn}_k(S)$. To acquire the valid learning signal for this highest-order feature, two strict conditions must be met simultaneously:
> > >
> > > 1. **Loss Generation:** At least one token must be masked to serve as the prediction target. (Masking zero tokens generates no loss).
> > >
> > > 2. **Context Preservation:** The remaining tokens must remain completely unmasked. By its definition in [4], pure synergy collapses to zero if even a single required context variable is missing.
> > >
> > > Consequently, masking multiple tokens destroys the synergistic context, while masking zero tokens provides no learning signal. The **only** valid configuration that yields a non-zero learning signal for the $k$-th order dependency is when exactly **one** token is masked (the target) and the remaining $k-1$ tokens are unmasked (the context). This exact boundary condition mathematically forces the learning dynamics of generalized $k$-gram to fall back directly to the parity case.
> > >
> > > [1] Prediction and entropy of printed English, Shannon, C.E., 1951.
> > >
> > > [2] Speech and Language Processing: An Introduction to Natural Language Processing, Computational Linguistics, and Speech Recognition with Language Models, 3rd edition.
> > >
> > > [3] Quantifying high-order interdependencies via multivariate extensions of the mutual information.
> > >
> > > [4] Nonnegative decomposition of multivariate information.
> > >
> > > [5] Sgd learning on neural networks: leap complexity and saddle-to-saddle dynamics.

---

### Decision · Program_Chairs · 2026-04-30

**Decision:**

Accept (regular)

**Comment:**

The paper studies Masked Diffusion Models (MDM) from a theoretical perspective, by analyzing the k-parity task.

This paper sparked a discussion among reviewers. The strength of the theoretical analysis and its insights has been praised. Furthermore, the k-parity experiments in controlled setting supports well the claims. A second part of the paper establish a link with language modeling, with both Nano-GPT experiments for pretraining from scratch, and SFT on LLaDA checkpoint.

However, a concern around the concept of "signal-optimal" terminology has been raised. It is important to replace it with "signal-rich" to avoid overstating the result. Another line of criticism is the link between k-parity task and realistic language modeling task: the link is mostly heuristic, and lack actionable insights for practitioners (Reviewer 7zJq, Reviewer y6GL).

Despite a lack of theoretical evidence on the link between k-parity and language modeling, the experimental part (LlaDA finetuning + NanoGPT pre-training) is a honest attempt at applying the theory at larger scale. Bridging theory and practice is always challenging in deep learning, as the structure of real datasets can be hard to capture with mathematical tools. Seeking intuition from theory and relying on experimental evidence when scaling-up is a valid scientific methodology. Especially because the improvements on ARC-Easy benchmark at pre-training time are not negligible.

Authors did a commendable job in the rebuttal to address some of the points raised by the reviewers. In particular, Reviewer y6GL brought some concerns around novelty, but failed to provide evidence of concurrent work, and did not answer the last message of the authors.

It must be noted that some of these insights transfer more broadly to other modalities (not only text), which can make the theoretical contributions of this paper of some interest to other tasks. Finally, authors provide a knob for practitioners to tune the amount of regularization in their training. Overall, I believe that these ideas are useful for community to build upon.

It is important that the authors remove all references to the "signal-optimal" terminology for the camera-ready version and replace it with the "signal-rich" instead. The section 5 should also be enriched with the rebuttal discussion, in particular the discussion with Reviewer k5bq and Reviewer 7zJq.